# High-quality and controllable time series generation with diffusion in transformers

## Abstract

Current research on time series generation frequently depends on oversimplified data and lenient evaluation methods, making it challenging to apply these models effectively in real-world scenarios. Diffusion in Transformers (DiT) has demonstrated that the traditional inductive biases in neural networks are unnecessary. This paper shows that the advantages of DiT can be extended to time series generation. We add the attention mask and dilated causal convolution to introduce the temporal characteristic. Additionally, we introduce a novel smooth guidance policy for style control during generation, leveraging a property of the diffusion process. Furthermore, our proposed model can generate longer sequences with training in short sequences. Experimental results reveal that our variant of DiT achieves state-of-the-art performance across various data types.

## 1 Introduction

Diffusion models Sohl-Dickstein et al. (2015); Ho et al. (2020); Nichol & Dhariwal (2021) have achieved remarkable results in image generation. Recent work Dhariwal & Nichol (2021); Nichol et al. (2021); Hatamizadeh et al. (2023); Hang et al. (2023) demonstrates that the generated images can capture features so convincingly that they are difficult to distinguish from real images by humans. Meanwhile, many other application areas are eager to benefit from the advancements of generative models, including finance, transportation, climate, medicine, etc. Reviewing the original intention behind generative models, the primary goal of generative research was to fit the original data distribution to enhance the generalization of specific task models Goodfellow et al. (2016; 2014). For instance, generating safety-critical scenarios Ding et al. (2023) is essential to improve the robustness of autonomous driving systems in dangerous situations. Another example is that Weber et al. (2008) train reinforcement learning agents in generated environments to reduce training costs. Although the most popular research continues to focus on image and language domains, the data types promoting industry development are predominantly time series.

On the other hand, Transformers Vaswani et al. (2017) and its derivatives Carion et al. (2020); Dosovitskiy et al. (2020); Beal et al. (2020); Zheng et al. (2021); Kirillov et al. (2023) have demonstrated that purely attention-based layers can replace traditional neural network architectures. From another perspective, the translation invariance of convolutional neural networks (CNNs) can be seen as an infinite strong prior Goodfellow et al. (2016), and this inductive bias is unnecessary. (Although the experiments in this paper show that this inductive bias accelerates convergence). Numerous studies have combined Transformers and ResNets He et al. (2016) in natural language processing Devlin et al. (2018); Ramesh et al. (2021), local image editing Hertz et al. (2022a), etc. Recently, Diffusion in Transformers (DiTs) Peebles & Xie (2023) successfully used Transformers as the backbone of a diffusion model, achieving state-of-the-art results in image generation. Naturally, we aim to adopt these breakthrough technologies to develop a flexible model framework for the time series field. This model should be suitable for complex and realistic generation tasks.

Although some works have successfully generated time series, three shortcomings have limited their practical applicability: 1) Generally, the generative model uses an autoregression-based backbone to introduce time series characteristics. The computations are usually sequential and cannot be fully parallelized. Furthermore, our experiments find that the too-strong temporal priors causes higher noise in generated samples. These noises or spikes can cause model collapses in dense time series data spaces. 2) There is a lack of effective conditional guidance strategies and model evaluation

methods. Most studies do not focus on conditional/style-guided time series generation and style transfer nor quantify diversity. Their metrics for evaluating generators typically use discriminators to distinguish real from fake data and predictive models to assess the correlation of time series in the time dimension. However, our experiments discuss the necessity of using classifiers to evaluate fidelity and diversity. 3) Real time series data cannot be scaled to uniform pixels like images. This is because the time interval is set to a fixed value, while the duration of events in the same dataset is usually different. At the same time, the underlying tasks require the generation of longer segments than the training data, such as stock and weather generation, which are trained in segments and generate samples lasting for many years. The custom methods of data synthesis are complex and may cause patterns lost.

Based on these shortcomings, This paper designs the diffusion in transformer for time series generation(timeDiT). We demonstrate that DiTs can be adapted for time series fields with simple and efficient modifications, with the proposed timeDiT model maintaining scaling properties and exploring the impact of introducing time priors on the model. We modify the diffusion process to generate feature-fused time series without additional model training.Our experiments are designed to evaluate pattern coverage capability, sample fidelity, and the practical usefulness of the generated data for low-level applications.

More specifically, the main contributions can be described as:

- We propose timeDiT, which introduces time characteristics based on dilated causal convolution, achieving performance far exceeding similar benchmarks across various indicators. Compared with similar diffusion-based models, it is more concise and efficient.

- We propose a method to fuse different categories of features in the diffusion step. Additionally, our model can accept training data of varying lengths and generate data more than ten times longer without distortion.These two are unique designs that consider the real application.

- For the first time, we employ classifier-based metrics in time series to assess model generation quality and ability to capture diversity, wheras previous work could only evaluate temporal characteristics.

## 2 RELATED WORK

**Time sequence** In this part, we not only discuss the generation of time series Yoon et al. (2019); Xu et al. (2020); Desai et al. (2021); Chen et al. (2020); Kong et al. (2020); Yuan & Qiao (2024), but also prediction and interpolation Tashiro et al. (2021); Zhou et al. (2021); Wu et al. (2021); Zeng et al. (2023); Zhou et al. (2022), with the latter two inspiring the representation learning of time series. The generator aims to capture the temporal relationships of all patterns and sample high-quality sequences. In score-based models, the data distribution will be concentrated on stronger peaks, whereas GAN-based models suffer from mode collapse, which affects the diversity of sampling. TimeGAN Yoon et al. (2019) ensures that the latent variable space retains temporal characteristics by training additional supervisors. Abhyuday proposed TimeVAE Desai et al. (2021), which provides an interpretable and fast training method. However, during the reproduction process, it was found that mixed patterns with significantly different characteristic peaks are difficult to capture simultaneously, requiring extensive hyperparameter tuning. Diffusion models have been successfully applied to time series generation in various works Chen et al. (2020); Kong et al. (2020); Yuan & Qiao (2024); Coletta et al. (2024); Alcaraz & Strodthoff (2022); Song & Ermon (2019), with Chen et al. (2020); Kong et al. (2020) using RNN as the backbone. In addition to the generation task above, most studies on time series concentrate on prediction tasks, including innovations in representation learning and decomposition of time series. Informer Zhou et al. (2021) demonstrates that transformers have strong representation capabilities for time sequences. Autoformer Wu et al. (2021) introduces Fourier transforms to guide decomposition tasks based on frequency. Spectral analysis is generally more widely used in audio signals, and Diffwave Kong et al. (2020) also uses the Mel Spectrogram of speech data as a conditional guide. As a unique time series attribute, frequency typically has different applications depending on the specific time task. For non-periodic, extremely low-frequency data in small windows, spectral analysis is limited.

**Diffusion Model**    The Denoising Probabilistic Model (DDPM) Ho et al. (2020) has made great achievements on image generation through the optimisation of: accelerated sampling Song et al. (2020) , variance prediction Nichol & Dhariwal (2021), guidance Dhariwal & Nichol (2021) , latent space Rombach et al. (2022). Furthermore, DiT Peebles & Xie (2023) demonstrated that U-Net's inductive bias is not necessary for diffusion models and use transformers backbone for the first time. Inspired by the work of DiT, we believe that the autoregressive design in the time series model discussed in the previous paragraph can be replaced by a concise and efficient attention layer. The latest work from Yuan & Qiao (2024) leverages full transformers to decompose time series into periodic signals, seasonal signals, and noise based on high amplitudes, generating high-quality samples. This decomposition is equivalent to introducing additional priors for the data, thereby accelerating convergence. Their experiments performed well in periodic data. Their disadvantage is that this decomposition affects the generation of the noise part. Compared to their work, our model only uses the encoder and discards the inductive bias brought by this decomposition.

**Guide and Edit**    Another crucial area is data editing, specifically the edited form of time sequences. This discussion covers two main types: overall guidance and style transfer, and partial modification of data. Extensive work Hertz et al. (2022b); Wang et al. (2023); Yang et al. (2023); Everaert et al. (2023) has successfully generated text-guided images, demonstrating that generated content can be controlled. Image style transfer Wang et al. (2023) shows that diffusion models can implicitly interpolate data points on the manifold, a task typically achieved through GAN interpolation Zhu et al. (2017); Karras et al. (2019). Hertz et al. Hertz et al. (2022b) propose a method for controlling images through partial modification by editing the attention map. Their work is based on the observation that the structure of generated data is determined at an early inversion step in diffusion models, with the remaining steps filling in details. While most discussions use cross-entropy control, experiments in Peebles & Xie (2023) find that conditional guidance based on Adaptive Layer Norm (AdaLN) produces higher-quality samples.

Research as early as 2017 Huang & Belongie (2017) showed that learned layer norm shift and scale can effectively and smoothly perform style editing. Numerous studies Li et al. (2017); Perez et al. (2018) have highlighted the potential of AdaLN, suggesting it can be more effective than cross-entropy. In time series, AdaLN offers a significant advantage: parameterized smooth control to generate samples, distinct from classifier-free condition parameters. In AI applications, many generation tasks require smooth control characteristics, such as generating emotions in language, the driving style of autonomous cars, and the adaptive behavior in reinforcement learning. These control objectives often need precise and smooth adjustments. Therefore, this article discusses the potential of AdaLN in time diffusion models, highlighting its ability to provide such smooth control.

## 3 TimeDiT

This section first briefly reviews the components adopted from DiT. Then, we use dilated causal convolutions Van Den Oord et al. (2016) to introduce temporal characteristics in transformers and explain the advantages of this method in model simplification, information processing efficiency, and long sequence generation applications. Finally, we generate the time series with different category features by adjusting the diffusion process.

### 3.1 Preliminaries

**DDPM**    We first briefly introduce Denoising Diffusion Probabilistic Models (DDPM), which operate by transforming a data distribution into a Gaussian noise distribution through a forward process (Noted as $q(x)$) and then sampling by reversing this transformation (Noted as $p(x)$). The forward process adds noise by a fixed noise schedule: $[\beta_1, \beta_2, ...\beta_t, ..., \beta_T]$ into $x_0$ over a series of steps $t$, transforming it into a noise-dominant state $x_t$. It can be rewritten as:

$$x_t = \sqrt{\alpha_t}x_0 + \sqrt{1 - \alpha_t}\epsilon_t, \quad \epsilon_t \sim \mathcal{N}(0, I), \tag{1}$$

where $\alpha_t$ is calculated from the noise schedule $[\beta]$ and $\epsilon_t$ represents the reparameterized Gaussian noise at the time step $t$.

The generation **problem statement** can be described as sampling noise data $x_T \in \mathbb{R}^{L \times D}$, where $L$ is sequence length and $D$ is dimension per time step, then reconstructing the original data step by

step from the reverse process by learning the conditional distribution:

$$p_\theta\left(x_{t-1} \mid x_t, c\right) = \mathcal{N}\left(\mu_\theta\left(x_t, c\right), \Sigma_\theta\left(x_t, c\right)\right), \tag{2}$$

where $\mu_\theta\left(x_t, c\right) = \frac{1}{\sqrt{d_t}}\left(x_t - \frac{\beta_t}{\sqrt{1-\tilde{d}_t}}\varepsilon_\theta\left(x_t, c\right)\right)$.

The model predicts $\epsilon_\theta$ and $\Sigma_\theta$ by given $x_t$ and $c$. By following Nichol & Dhariwal (2021), $\epsilon_\theta$ is trained by:

$$L_{simple} = \mathbb{E}_{t,x_0,\varepsilon,c}\left[\|\varepsilon - \varepsilon_\theta\left(x_t, c\right)\|^2\right]. \tag{3}$$

Then $\Sigma_\theta$ is trained by: $\lambda L_{\text{vlb}} = \sum_t D_{KL}\left(q\left(x_{t-1} \mid x_t, x_0\right) \| p\left(x_{t-1} \mid x_t\right)\right)$, where $\lambda$ is scaling parameter.

In sampling, we follow classifier-free guidance Ho & Salimans (2022), that sampling $\tilde{\epsilon}_\theta\left(x_t, c\right) = \epsilon_\theta\left(x_t, null\right) + s \cdot \left(\epsilon_\theta\left(x_t, c\right) - \epsilon_\theta\left(x_t, null\right)\right)$, where $s$ is a scale factor that adjusts the influence of condition $c$ on the generation process.

**AdaLN**  DiTs find that the block with the adaptive layer norm initialised at zero (AdaLN-zero) performs best. Here, we follow this setting and briefly review it. The conditional information is slowly added by a layer in the $i$-th block: $\text{AdaLN}(x; i) = \gamma_i \cdot \text{LayerNorm}(x) + \beta_i$, where $\gamma_i$ and $\beta_i$ are learned scale and shift, obtained from a function approximator. Here we use a simple Multilayer Perceptron (MLP): $\gamma_i, \beta_i = \text{MLP}(c)$. Remarkably, this allows the network to generate sequences in various styles using the same model but in different diffusion steps and conditions.

## 3.2 DESIGN SPACE

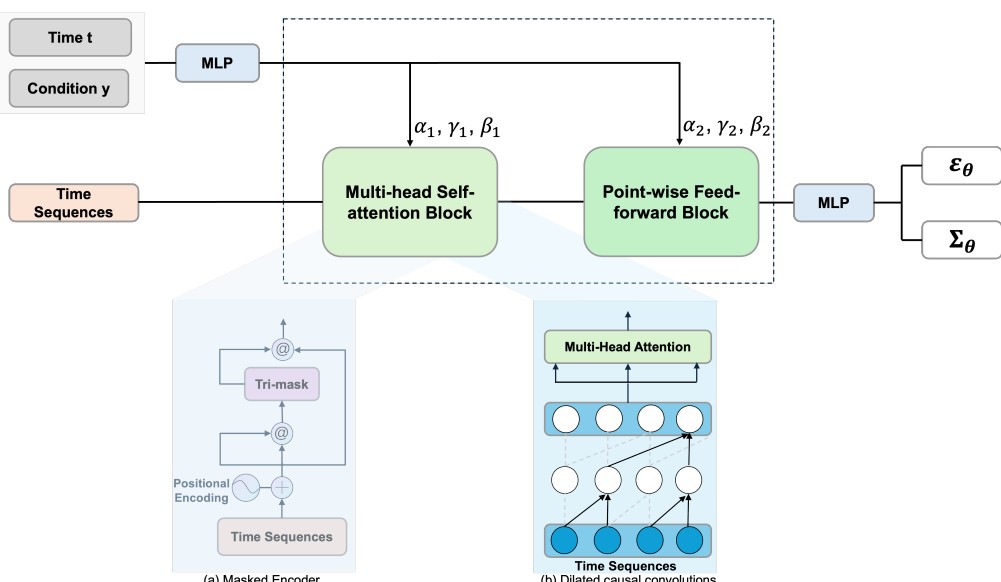

Figure 1: (a): Masking is used to prevent the current time step from accessing future time step information, which is common in the transformer's decoder. Using this masking in the encoder introduces temporal constraints. @ is matrix multiplication. (b): Dilated causal convolution layers modify the receptive field of the current time step, introducing temporal characteristics at a lower cost.

### 3.2.1 TIME PRIOR

**Masked Encoder**  An important characteristic of time series is that the data at the current time step can be obtained only from past time steps, without future data. Next, we will explain how to introduce this characteristic into the transformer. Referring to the use of position masks in the translation task

to mask future targets for parallel training, naturally, the lower triangular mask can be used in the self-attention layer in the encoder to mask the information of future time steps. As shown in Figure 1(a). Specifically, the strictly upper triangular part of the weight matrix of the self-attention layer is set to 0. Here $Output_i = \sum_{j=1}^{i} Weight_{ij} \cdot V_j$, where $Weight_{ij} = input_i \cdot input_j$. The output at length index $i$ is independent of input $i+1$ to $L$. This feature is still retained after passing the next layer of blocks.

**Soft prior** One concern is that the model's goal is to predict noise. Simply adding temporal characteristics to the noise scale will affect the generative capability. We find an interesting phenomenon that samples keep the features of the data but have more peaks (Figure 2a). In high-density time series data distribution, this often also results in the disappearance of certain patterns. Figure 2b shows the impact of training on a noisy scale that increases the sample noise. Salimans & Ho (2022) deduces that predicting $\epsilon_t$ is equivalent to multiplying the signal-to-noise ratio before the loss of predicting $x_t$. Figure 2b shows that predicting $x_{t-1}$ will lead to unstable training, which is more obvious in time series compared to image generation tasks. Therefore, it is vital to retain the advantages of prediction noise and variance while introducing temporal characteristics.

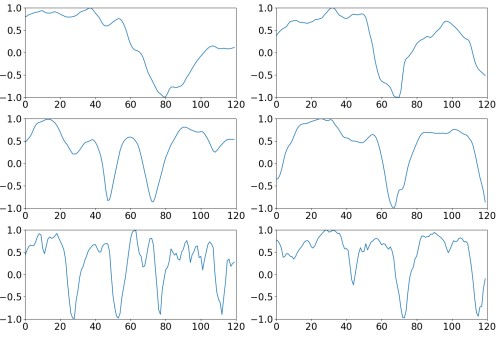 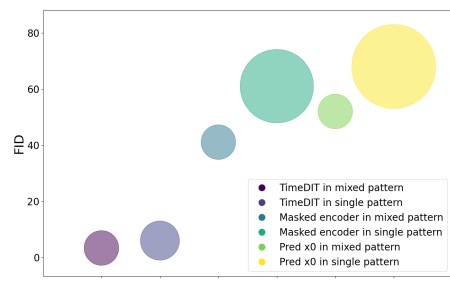

(a) Masking and positional encoding increase noise and the number of peaks

(b) Comparison of using masking, predicting $x_0$ with optimal settings

Figure 2: (a) demonstrates the drawbacks of introducing temporal priors using masking. Each column represents an example: the first row shows the real data, the second row shows data generated by timeDiT (using dilated causal convolution layers), and the third row shows samples from timeDiT based on masking. It can be observed that the third row contains more noise, which gets amplified during the diffusion steps, eventually forming additional peaks. (b) predicts noise and variance, which is better than directly predicting $x_0$, and the noise introduced by masking increases the FID value and contrast.

On the other hand, from the perspective of deep learning, too much price has been paid to introduce these temporal characteristics. First, masking causes half of the attention weights to be discarded. To ensure model capacity, depth and dimensionality need to be increased. Second, even with smaller scales, positional encoding introduces noise.

One solution is to introduce soft time prior (Figure 1(b). Dilated causal convolutional networks Van Den Oord et al. (2016) re-encode the time series before entering the multi-head attention layer so that the current time step contains all the receptive fields of the previous time steps. After entering the self-attention layer, the values of this time step are naturally weighted and added. This optimization avoids wasted attention weights and does not require position encoding. Since this approach preserves the connection with future time steps while making the current time step strongly correlated with past values, it becomes soft prior knowledge. This is reasonable in generative tasks (not prediction tasks). In the ablation study, we demonstrate the advantage of temporal priors introduced with dilated causal convolution.

**Longer sequences generation** We have removed the positional encoding from the Transformer and represented the data at each time step as a weighted sum of all previous time steps. The benefit of this improvement is not only to reduce the network size but also to make long sequence generation

available. Since there is no fixed position encoding and the convolution operation is based on a sliding window, during the sampling process, $x_t$ in the diffusion step can be a sequence of indefinite length. Subsequently, the output layer of the transformer should be designed as a **point-wise layer**. The point-wise layer is independent of the sequence length, allowing for training with time series of different lengths and generating time series of different lengths during sampling. This design is crucial because many applications that use time series data incur high costs to collect long sequences, so only short sequence data is typically available. Compared to autoregressive long sequence generators like decoders or RNNs, the method of expanding the receptive field with dilated causal convolutions allows for parallel generation.

**Smooth Control** An important phenomenon was observed in the work of Hertz et al. (2022b), in which the diffusion model generates the overall framework first and then the details. Additionally, Coletta et al. (2024) fixes the value of certain points in the diffusion process in $x_0$ to generate a time series that satisfies the constraints. Inspired by these two studies, we propose a novel method that samples data from a fused condition. Specifically, we generate the overall framework of the time series through the first $T - \tau$ steps of the diffusion step, with modifications to the shift and scale steps to guide the generation, as illustrated in Figure 3. By modifying the hyperparameter $\tau$ and replacing or interpolating the shift and scale values, the data can be guided to a controllable range:

$$\alpha, \beta, \gamma = \begin{cases} MLP(Embed\,(t) + Embed(y)), & t < \tau \\ MLP(Embed\,(t) + Embed(y')), & t \geq \tau \end{cases}, \quad (4)$$

where $\alpha$, $\beta$ and $\gamma$ represent all scale and shift values. The $y$ and $y'$ are the condition labels aim to infuse.

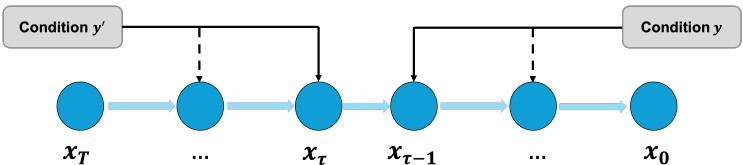

Figure 3: Hertz et al. (2022b) store a weight map in the buff to complete partly edit. In style infuse, This algorithm can be simplified as figure shown because the condition is introduced by AdaLN instead of cross-attention.

## 4 EXPERIMENTS

In Section 4.1, we describe the experimental data, benchmarks, and adopted metrics. In Section 4.2 we design the comparative experiments to show the superiority of time DiT over related work. The experiments in Section 4.3 demonstrate the effectiveness of the proposed style control method and evaluate the performance of generating sequences longer than the training data. In the field of time series, this model is the only one that can accomplish these two underlying tasks. Finally, in the ablation experiments in 4.4, we replace different designs to demonstrate the effectiveness and superiority of introducing temporal features with dilated causal convolutions. In addition, we put some important experiments in the appendix, including the impact of classifier error on evaluation (Appendix B.3), the scaling characteristics of timeDiT (Appendix B.4 hyperparameter), visualization of pattern coverage, and additional generation results. All of the models were **not** fine-tuned, and all the samples were randomly chosen, **not** selected.

### 4.1 SET UP

**Dataset** Our experimental data includes driving cycle Oh et al. (2020), stock, weather, solar, and traffic trajectory data Wilson et al. (2023) with segment lengths of 120 steps. A sequence length of 120 is chosen to capture sufficient data characteristics and meet practical needs across various fields. The selected data addresses popular applications and diverse time series characteristics. For example, the driving cycle sampled at 10 Hz is typically flatter with fewer peaks. More details on data processing and experimental design are available in Appendix A.1 - A.2.

**Metrics** Evaluating generative models with discriminant and prediction scores alone is insufficient, especially for conditional generation tasks. These metrics can't ensure all modes are captured, impacting diversity. Even with adequate timing information, quality may be poor. Inspired by image generation, we introduce a classifier Ismail Fawaz et al. (2020) based on a 1-D convolution network to evaluate IS and FID for time series (details in Appendix A.3). Although not perfect and influenced by classifier performance, these metrics provide relative evaluation quality. Specific physical constraints should be considered at the application layer, beyond this article's scope. Additionally, metrics similar to classifier accuracy and recall are introduced for condition generation, differing from those in Sajjadi et al. (2018).

### 4.2 GENERATOR EVALUATION

**Unconditional generation** Table 1 compares the performance of timeDiT with the baseline on various tasks. The first four evaluations are for single data types, while mixed data includes all four types, representing a mixed-density distribution with distant peaks. Under single data, TimeDiT outperforms TimeGAN and TimeVAE, and performs comparable to the diffusion model DiffTS with high decomposition prior. Under multimodal mixed data, TimeDiT achieves leading fitting indicators. In such tasks, timeGAN and timeVAE struggle to separate patterns, as seen in the IS scores where they lose a class. For single data sets, the driving cycle isn't well represented by timeGAN and timeVAE due to slight noise being mistaken for weather data. The periodic decomposition assumption of DiffTS is not conducive to the modal fitting of mixed data. We additionally compared the fitting effects of timeDiT and a similar diffusion model DiffTS using periodic decomposition and Fourier loss on mixed data (Figure 4), and found that after sufficient training time, timeDiT performs much better than DiffTS. By comparing the generation results of TimeDiT and other models (Appendix C.4), it is found that the generation curve of timeDiT is always smoother, which indicates that its noise is significantly lower.

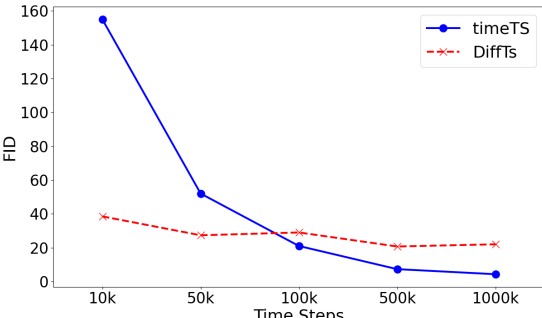

Figure 4: Comparative experiment of DiffTs and timeDiT.

Figure 4 shows the performance changes of DiffTs and timeDiT as the training steps increase. The disentangled prior introduced by DiffTs brings faster convergence, but its prediction of $x_t - 1$ and the setting of Fourier loss actually reduce the performance after convergence. The scaling properties of timeDiT start to bring significant advantages after $100K$ training steps, proving that this prior is unnecessary.

**Conditional generation** Table 2 records the experimental results of conditional generation. Compared to the FID of unconditional models trained on single datasets, conditional generation produced data without distortion. The accuracy and recall values are means and variances from 20 independent experiments, showing that timeeDiT can perfectly generate data of specified categories.

**Correlation constraints on multivariate sequences** Another experiment is designed to demonstrate the model's understanding of multivariate time series. Table 3 evaluates the interrelationships between variables, using MSE to assess the physical consistency of driving trajectory and speed components. Table 3 presents the sample quality assessment, showing that timeDiT performs best on

Table 1: Comparison table of unconditional generation results. The chosen benchmarks are respectively based on GAN, VAE, and the most advanced diffusion-based models. Diffwave and DiffTS are based on full convolution and transformer decoder autoregression respectively. Since the single data set has only one category, we use the average entropy of classification to replace the IS value. The lower the entropy, the higher the confidence that the data is recognized as a certain category, that is, the generated data is better. Bold indicates the best model for the current sub-experiment.

| Dataset | Model | Metrics | | | |
|---|---|---|---|---|---|
| | | IS↑/Entropy↓ | FID↓ | Discriminative Score↓ | Predictive Score↓ |
| Driving cycle | TimeDiT | 0.007 | 3.24 | 0.153±.090 | **0.192±.000** |
| | Diffwave | 0.105 | 5.45 | 0.274±.052 | 0.244±.002 |
| | Diffusion-TS | **0.002** | **1.50** | **0.051±.076** | 0.193±.000 |
| | TimeGAN | 0.164 | 39.72 | 0.246±.038 | .255±.003 |
| | TimeVAE | 0.19 | 27.74 | 0.299±.105 | .254±.002 |
| Stock | TimeDiT | **0.002** | 9.06 | **0.150±.057** | 0.249±.004 |
| | Diffwave | 0.006 | 11.45 | 0.467±.064 | 0.297±.001 |
| | Diffusion-TS | 0.006 | **5.44** | 0.193±.087 | **0.195±.000** |
| | TimeGAN | 0.049 | 10.27 | 0.569±.028 | 0.260±.000 |
| | TimeVAE | 0.009 | 11.02 | 0.525±.031 | 0.263±.000 |
| Weather | TimeDiT | **0.002** | **6.09** | **0.158±.068** | **0.249±.006** |
| | Diffwave | 0.003 | 8.60 | 0.299±009 | 0.299±.004 |
| | Diffusion-TS | 0.006 | 11.00 | 0.275±.004 | 0.254±.000 |
| | TimeGAN | 0.008 | 9.14 | 0.319±.144 | 0.288±.000 |
| | TimeVAE | 0.005 | 8.86 | 0.482±.010 | 0.265±.002 |
| Solar | TimeDiT | **0.000** | **3.54** | **0.247±.105** | **0.238±.001** |
| | Diffwave | **0.000** | 4.08 | 0.400±.005 | 0.255±.001 |
| | Diffusion-TS | **0.000** | 4.31 | 0.290±.025 | 0.264±.000 |
| | TimeGAN | 0.002 | 4.04 | 0.428±.003 | 0.247±.004 |
| | TimeVAE | 0.002 | 4.40 | 0.430±.001 | 0.258±.005 |
| Mixed data | TimeDiT | **3.98** | **3.96** | **0.118±.065** | **0.274±.002** |
| | Diffwave | 3.27 | 18.54 | 0.255±.138 | 0.292±.001 |
| | Diffusion-TS | 3.75 | 12.60 | 0.395±.057 | 0.285±.001 |
| | TimeGAN | 1.913 | 45.56 | 0.499±.001 | 0.461±.015 |
| | TimeVAE | 2.27 | 35.60 | 0.498±0.002 | 0.404±.055 |

Table 2: Conditional generation results

| Dataset | Metrics | | |
|---|---|---|---|
| | FID | Precision | Recall |
| Driving cycle | 3.22 | 1.000 | 0.997 |
| Stock | 9.43 | 0.999 | 1.000 |
| Weather | 7.37 | 0.997 | 0.999 |
| Solar | 3.06 | 1.000 | 1.000 |

Table 3: Comparative results on multivariate task

| Model | Metrics | | |
|---|---|---|---|
| | IS | FID | MSE |
| TimeDiT | **2.900** | **1.63** | 0.0206 |
| Diffusion-TS | 1.63 | 14.7 | **0.0179** |
| TimeGAN | 1.37 | 59.02 | 0.0501 |
| TimeVAE | 1.54 | 35.6 | 0.0396 |

*Note:* Original data has MSE=0.003 basic error.

multivariate time series. Notably, DiffTs excels in real physical descriptions due to its advantage in time series decomposition.

### 4.3 EXTENDED EXPERIMENTS

**Smooth controllability** In this section, we present the results of controllable generation using different $\tau$ values in the diffusion model. Figure 5a shows a speed curve with low values at both ends and high values in the middle, generated by gradually reducing the weight of the solar label in the

diffusion process. Figure 5b displays a speed curve with stock noise, produced by incorporating stock diffusion guidance into the generation process. It can be found in Table 4 that, even if only the last 50 time steps are used for driving label guidance, the generated data retains enough features of the driving data. (FID is around 23)

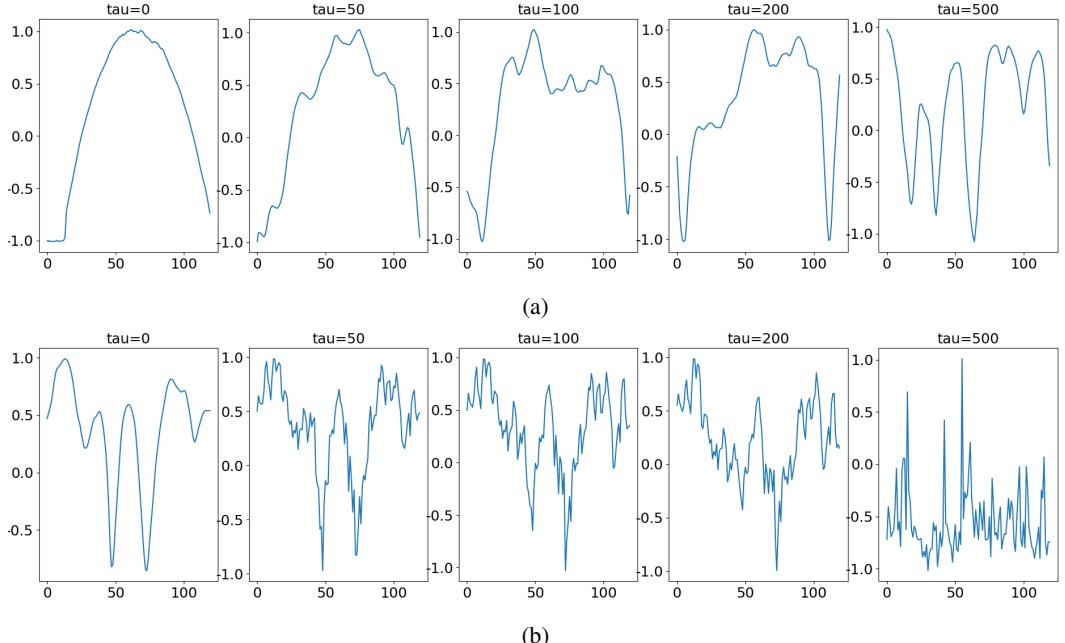

(a)

(b)

Figure 5: With different proportions of labels in the diffusion step, the generated data presents different ratios of feature fusion

Table 4: Generative data evaluation from different $\tau$.

| $\tau$ | 0 | 50 | 100 | 200 | 500 | 1000 |
|---|---|---|---|---|---|---|
| $FID_a$ | 171.53 | 21.21 | 11.26 | 10.34 | 4.75 | 4.02 |
| $FID_b$ | 245.36 | 25.29 | 15.87 | 12.88 | 12.76 | 4.7 |

**Longer sequence generation**  Table 5 demonstrates the application of generative models for long sequences. The experiment trained on data with $Length = 120$ and generated samples of $Length = 1200$ without preset settings. This is crucial for practical applications where only segmented data can be sampled due to cost constraints, such as urban traffic trajectories or energy life cycles. Extended generation can also be combined with style-controlled generation for varying multimodal sequences. Results in Table 5 show that extended sequences have slight distortions in small segments but outperform the baseline.

Table 5: Different long sequences results

| $Length$ | 120 | 240 | 360 | 480 | 1200 | 2400 |
|---|---|---|---|---|---|---|
| $FID$ | 3.44 | 15.86 | 15.62 | 19.53 | 21.18 | 37.18 |

## 4.4 ABLATION STUDY

In this section, we present the ablation experiments on both single and mixed datasets. The results in Table 6 show that DiT, which discards temporal characteristics, lacks the ability to fit time curves. The reason is that in the design of DiT, data at different time steps are independent of each other. DiT with positional encoding and temporal masking performs well on noisy data but fails to generate

high-quality smooth data. This defect leads to the disappearance of smooth velocity curve patterns. TimeDiT, which introduces dilated causal convolution, performs well across various datasets. We show the sampling of different components in the appendix, where TimeDiT can generate high-quality samples without noise.

Table 6: Ablation study results ( $TimeDiT$: $TimeDiT$ with dilated causal convolution; m_with_pos: Mask with positional encoding; m_w/o_pos: Mask without positional encoding; w/o_AdaLN: Unconditional generation without AdaLN; with CNN: replace DCC by 1D-CNN)

| Matrics | IS | FID | Avg_Precision | Avg_Recall |
|---|---|---|---|---|
| $TimeDiT$ | 3.98 | 3.96 | 0.999 | 0.999 |
| m_with_pos | 3.72 | 25.29 | 0.923 | 0.912 |
| m_w/o_pos | 0.93 | 243.76 | 0.275 | 0.249 |
| with CNN | 1.34 | 157.23 | 0.348 | 0.292 |
| w/o_AdaLN | 3.23 | 45.19 | 0.858 | 0.821 |

## 5 LIMITATION

Firstly, there is a lack of a unified time series dataset for consistent comparison of models, as real-world data varies greatly, making cross-dataset evaluation challenging. This paper details the rationale for classifier-based evaluation metrics. Secondly, while timeDiT achieves leading results, it requires the longest training time. DiffTs can generate low-noise data in 10k steps, and timeVAE converges in 1k steps, raising considerations about trading training time for quality. In fact, due to the need for sampling across diffusion steps T, diffusion models typically require over 100K training steps to ensure sufficient coverage at each time step. However, diffusion models based on transformers tend to be less sensitive to hyperparameters compared to GANs and VAEs, making the training process easier to converge. Lastly, a common limitation of diffusion models, timeDiT has the longest sampling time. Appendix shows accelerated sampling with DDIM, yet timeDiT's sampling time remains higher than timeVAE and timeGAN.

## 6 CONCLUSION

This paper enhances the receptive field of Transformers by extending causal convolution, allowing each time step to be a weighted sum of previous steps. This soft temporal prior eliminates the need for positional encoding and temporal masking, improving the model's understanding. Our model surpasses benchmarks in modal capture ability and generation quality. Additionally, our research shows that timeDiT retains scaling properties in time series generation and captures more multivariate sequence relationships. Finally, among similar studies, TimeDiT is the only model capable of scaling to controllable conditional fusion and the generation of longer sequences, demonstrating its effectiveness in practical applications.

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

### AUTHOR CONTRIBUTIONS

If you'd like to, you may include a section for author contributions as is done in many journals. This is optional and at the discretion of the authors.

### ACKNOWLEDGMENTS

Use unnumbered third level headings for the acknowledgments. All acknowledgments, including those to funding agencies, go at the end of the paper.

## A    APPENDIX

## B    IMPLEMENTATION DETAILS

### B.1    DATA PROCESS AND EXPERIMENT DESIGN

To enhance the applicability of the generated data, we meticulously designed challenging experiments. Table 7 presents all the datasets used in this paper. The **driving cycle dataset** represents speed over time, with a time interval of 0.1 seconds. Consequently, its temporal characteristics are relatively smooth curves, and due to acceleration limits, there are no excessively steep peaks. Moreover, the number of peaks over the entire 120-length sequence should be relatively low. The **stocks dataset** comprises manually downloaded historical records of over 100 listed companies, including daily high prices, low prices, and trading volumes, with a time interval of one day. The **weather dataset** includes daily atmospheric pressure, temperature, and humidity, with a time interval of one day. The **solar dataset** contains the total power of regional users, with a time interval of 12 minutes. We split each dimension into 1-dimensional time series because our experiment design in this section focuses more on data diversity and generation quality rather than representation learning.

Stock data exhibits high volatility, weather data shows overall stability with local fluctuations, and solar data peaks are concentrated in the middle (higher daytime electricity usage). Therefore, we selected datasets that cover a wide range of time series characteristics, each with distinct features. In **mixed data**, we combined the datasets to test the model's ability to capture all patterns.

For recognizing and generating high-quality multivariate time series, we used the **Argo2 dataset**, a 5-dimensional time series $[pos_x, pos_y, heading, v_x, v_y]$, where the next moment's position is strongly related to the current five data points. We demonstrate that TimeDiT's capability to understand these data without any prior knowledge.

Table 7: Datasets

| Dataset | Samples | Link |
|---------|---------|------|
| Driving cycle | 85057 | `https://github.com/gsoh/VED` |
| Stock | 10567 | `https://finance.yahoo.com/quote/GOOG/history` |
| Weather | 23354 | `https://www.bgcjena.mpg.de/wetter/weather_data.html` |
| Solar | 12307 | `https://www.nrel.gov/grid/solar-power-data.html` |
| Argo2 | 300k | `https://www.argoverse.org/av2.html#download-link` |

## B.2 METRICS

Our design incorporates classifier-based metrics (IS and FID. Previous work utilized discriminative scores and predictive scores to evaluate the generated time series. However, these evaluation scores do not aid in assessing conditional guidance and pattern coverage. Although t-SNE can be used to project data onto a 2D coordinate system for coverage visualization, this method lacks quantitative metrics. Furthermore, data with good discriminative and predictive scores may still be suboptimal. For example, in our experiments, when the generated driving cycle was subjected to excessive noise resulting in numerous small peaks, the data still maintained good discriminative and temporal characteristics scores. However, the FID value significantly deviated from that of all data types, indicating that such data is unacceptable.

## B.3 IMPERFECT CLASSIFIER ANALYSIS

In image generation, generators are evaluated on the same dataset and with the same classifier. However, this consistency cannot be guaranteed in time series generation. A wide range of lower-level applications require different types of time series, which is one reason why previous experiments did not use classifiers. Nonetheless, we still need classifiers to identify the correct patterns. For unlabeled data, classifiers can be replaced with arbitrary feature extractors to calculate FID values.

One concern is whether the evaluation method in this paper is reliable. In Figure 6, we discuss the impact of imperfect classifiers on experimental results. In Figure 6a, we examine the evaluation capability of different classifiers on the model. The conclusion is that classifiers performing well on the test set consistently retain relative model quality differences. In other words, different classifiers may cause slight variations in FID values, but a generative model that performs well under one classifier will not perform poorly under another. This is consistent with theory because differefigurent classifiers have varying representation capabilities, but a good representation model consistently reflects the quality of the generative model. In Figure 5b, we observe the impact of underfitting classifiers on the evaluation of generative models.

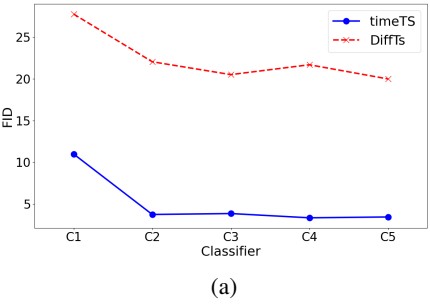
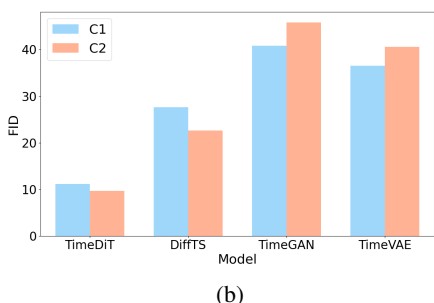

(a)                  (b)

Figure 6: Impact of imperfect classifiers on experimental results.

## B.4 HYPERPARAMETER

All tensor calculations are running in RTX 3080 with 10GB memory. The idea of training memories should be more than 2GB. Sampling #8000 data needs a separate 2GB memory. This section discusses the characteristics of Transformers in time series generation. Table 8 shows the impact of different hyperparameter selections on model evaluation. The first row displays the optimal model design. Firstly, a learning rate of $1 \times 10^{-4}$ and a batch size of 32 provide the most stable training setup. Reducing the learning rate does not significantly improve the model. Secondly, the optimal depth and dimension are 6 and 16, respectively. Reducing this depth

or dimension significantly degrades the quality of the generated model. Increasing the dimension beyond 16 markedly enhances the generative capability but results in a substantial increase in model parameters. Increasing the depth is unnecessary because, for the designed experiments, the improvements brought by increased depth do not outweigh the memory and computational costs.

Combining the experiments on highly correlated multivariate sequences discussed in the main text, our conclusion is that the model should be designed according to the specific generative task. A dimension of 16 is the optimal setting for representing 1-dimensional sequences. To capture finer modal differences, increasing the depth may be required.

Table 8: Performance under different model capacities and different settings.

| Parameter | Depth | Dimensions | Attention Heads | Batch Size | Learning Rate | FID | Training Steps |
|-----------|-------|------------|-----------------|------------|---------------|-----|----------------|
| 39k | 6 | 16 | 4 | 32 | $1 \times 10^{-4}$ | 3.36 | 1000k |
| 49k | 8 | 16 | 4 | 32 | $1 \times 10^{-4}$ | 3.34 | 2200k |
| 68k | 12 | 16 | 4 | 32 | $1 \times 10^{-4}$ | 3.75 | 2200k |
| 25k | 3 | 16 | 4 | 32 | $1 \times 10^{-4}$ | 10.08 | 3000k |
| 39k | 6 | 16 | 4 | 32 | $1 \times 10^{-5}$ | 3.83 | 2500k |
| 147k | 6 | 32 | 4 | 32 | $1 \times 10^{-4}$ | 2.77 | 2000k |
| 147k | 6 | 32 | 8 | 32 | $1 \times 10^{-4}$ | 2.75 | 2000k |
| 11k | 6 | 8 | 4 | 32 | $1 \times 10^{-4}$ | 31.98 | 1000k |
| 25k | 3 | 16 | 4 | 32 | $1 \times 10^{-4}$ | 8.00 | 1000k |

## C  ADDITIONAL RESULTS

In this section, we present additional results. In Section B.1, we use the t-SNE tool to visualize pattern coverage. Even without quantitative intuitive metrics, t-SNE can still reveal deficiencies in the model's fit for certain data. In Section B.2, we provide more samples generated through conditional fusion. In Section B.3, we show additional samples and metrics for long sequence generation.

### C.1  VISUALIZATION OF PATTERN COVERAGE

In Section B.1 from Figure 7 to Figure 11, We demonstrate the use of t-SNE and PCA to project generated and raw data into 2D plots to visualize pattern coverage.

### C.2  CONTROLLABLE CONDITIONS GUIDANCE

Here we show more controllable generated results from Figure 12 to Figure 13. Compared with replacing cross attention, the shift and scale values generated by the replacement condition change the sample style more generally rather than locally modifying it. This is consistent with time series application scenarios. Examples of applicable scenarios for time series style transfer include voice speaker replacement, driving aggressiveness, stock rises and falls, etc.

### C.3  LONG SEQUENCE GENERATION

We demonstrate the generation of sequences of length 480 (5L) and length 1200 (10L). Our results in Figure 14 show that when timeDiT generates longer sequences, it does not simply extend the original length, but retains the characteristics of the original data in all windows on the timeline.

### C.4  ADDITIONAL SAMPLES

Finally, we show additional generated samples and raw data samples from Figure 15 to Figure 18. The generated data retains the characteristics of the original data and is nearly indistinguishable to humans.

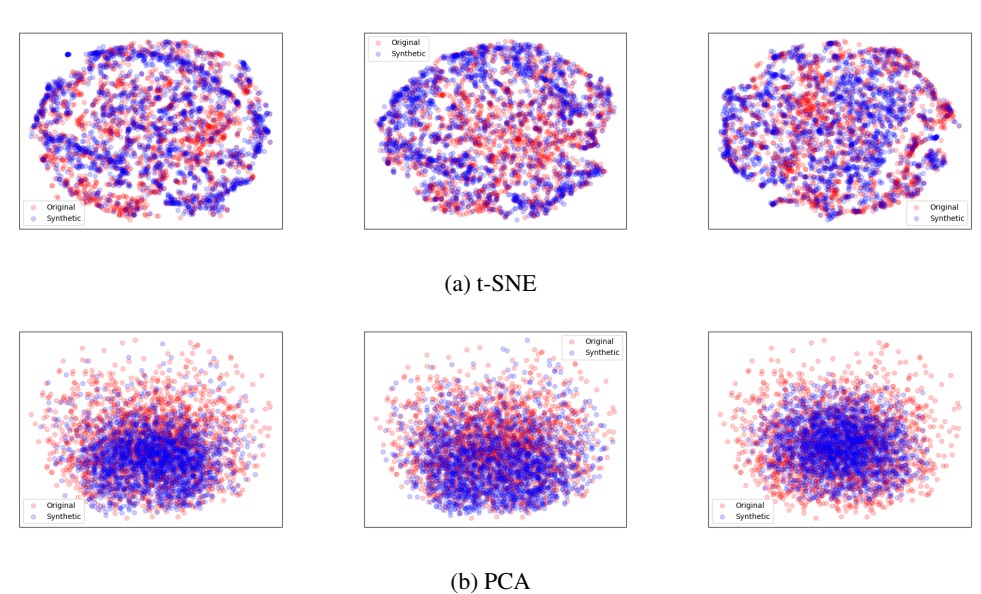

(a) t-SNE

(b) PCA

Figure 7: Visualization in driving cycle dataset. From left to right they are timeDiT, DiffTs, timeGAN

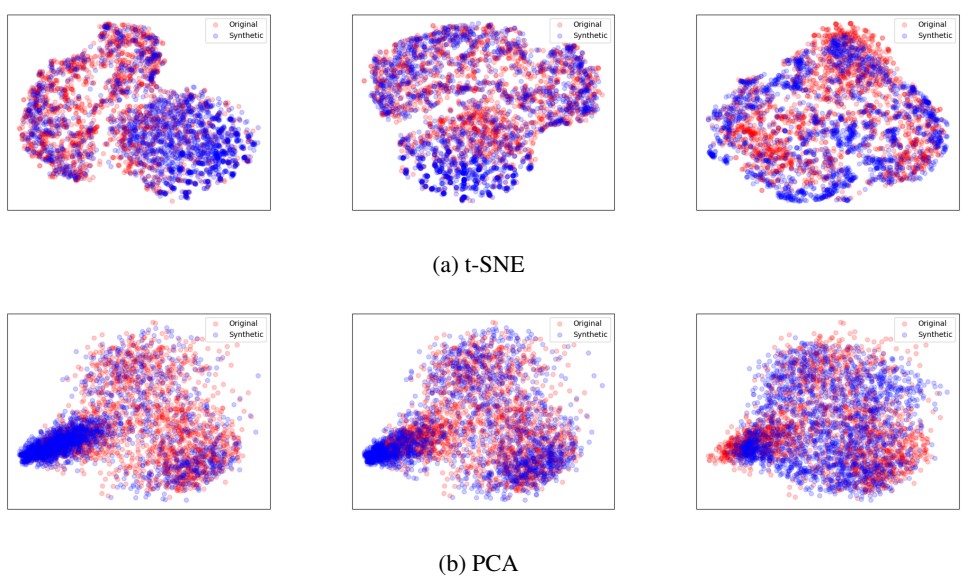

(a) t-SNE

(b) PCA

Figure 8: Visualization in stock dataset. From left to right: timeDiT, DiffTs, timeGAN

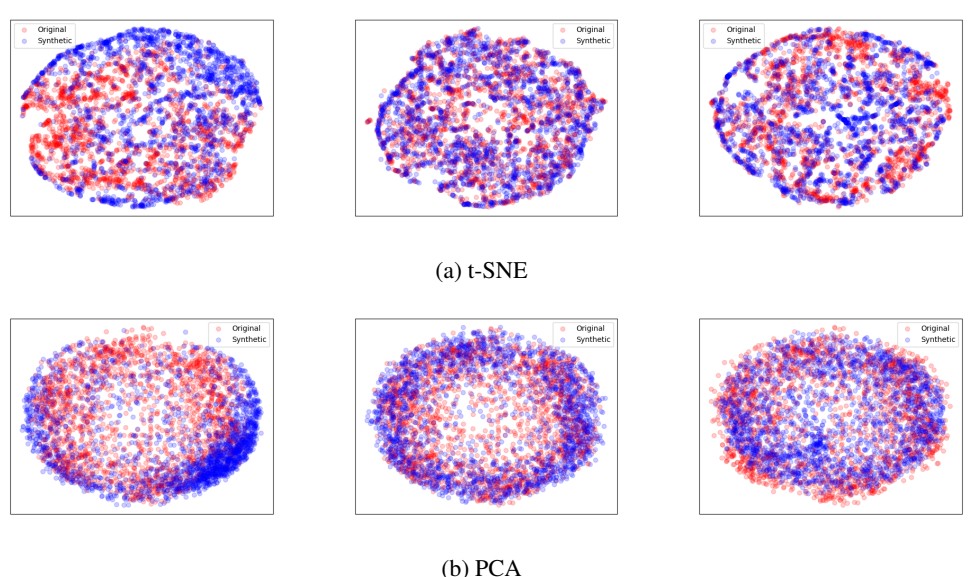

(a) t-SNE

(b) PCA

Figure 9: Visualization in weather dataset. From left to right they are timeDiT, DiffTS, timeGAN

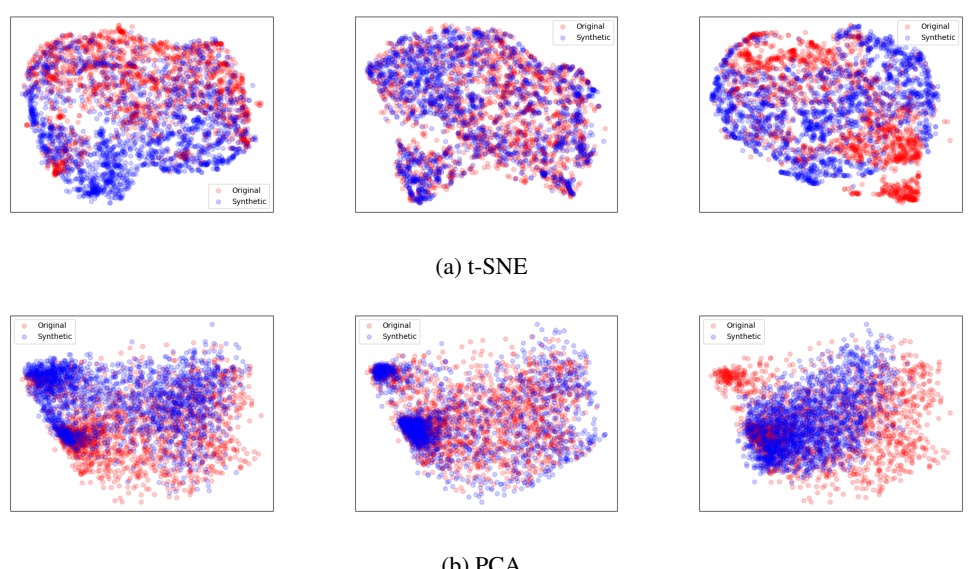

(a) t-SNE

(b) PCA

Figure 10: Visualization in solar dataset. From left to right they are timeDiT, DiffTS, timeGAN

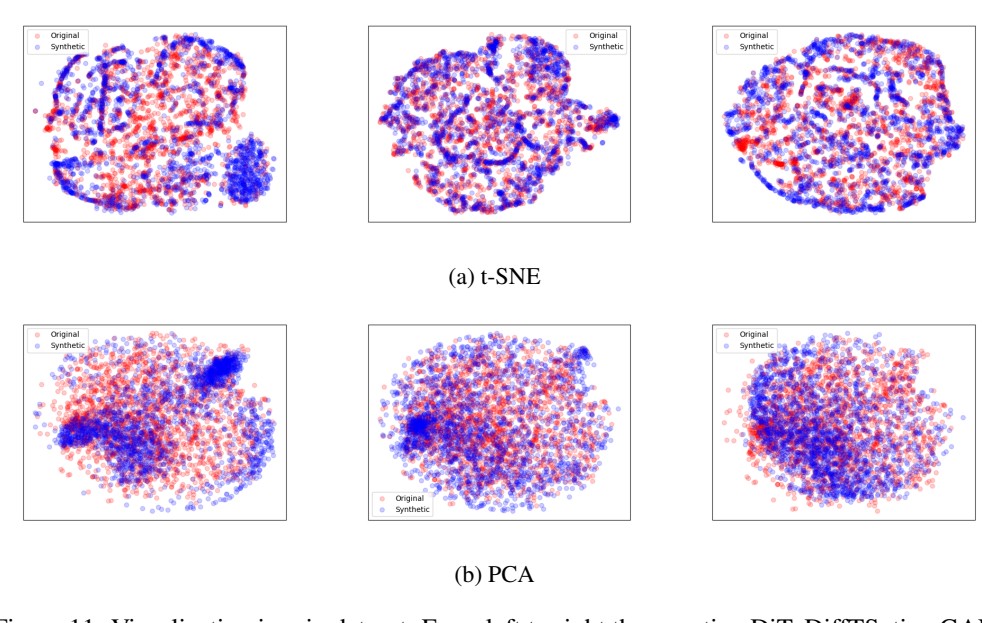

(a) t-SNE

(b) PCA

Figure 11: Visualization in mix dataset. From left to right they are timeDiT, DiffTS, timeGAN

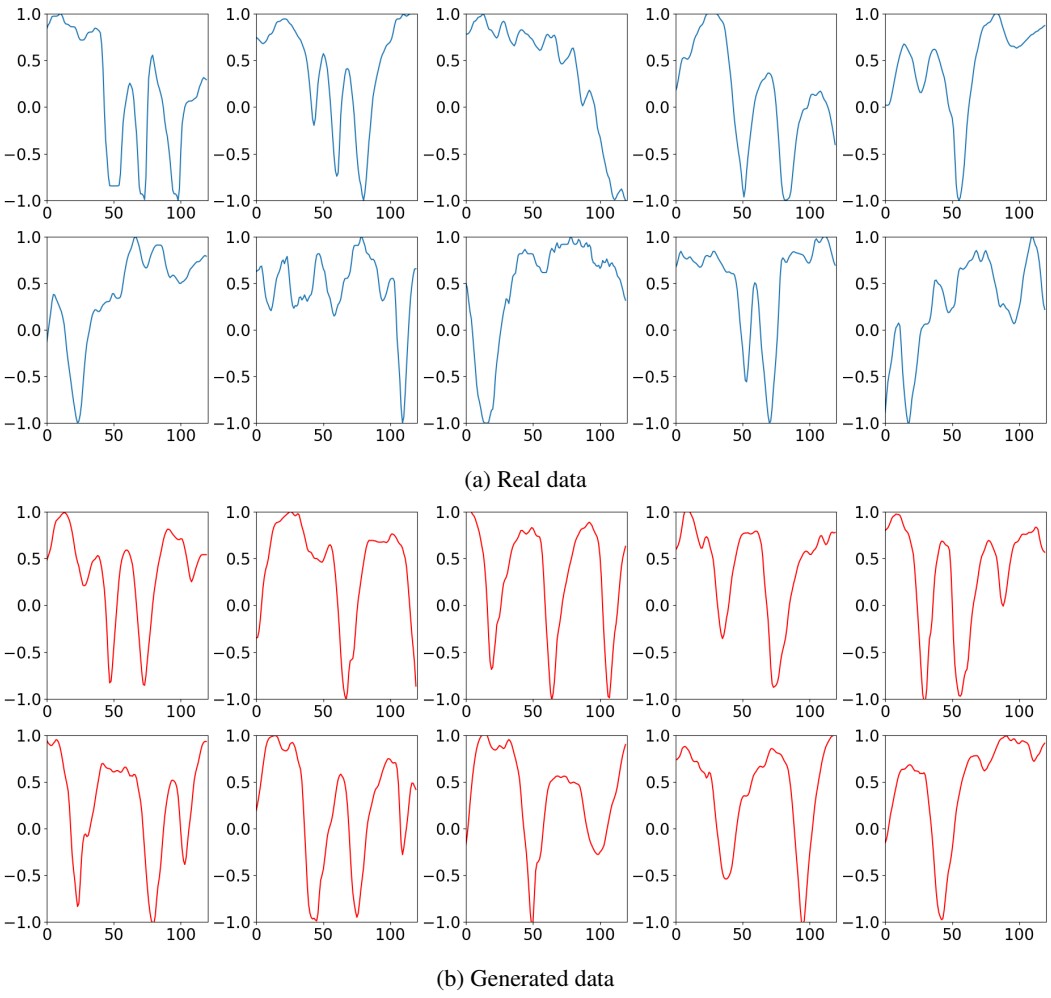

(a) Real data

(b) Generated data

Figure 12: Additional results for sample display

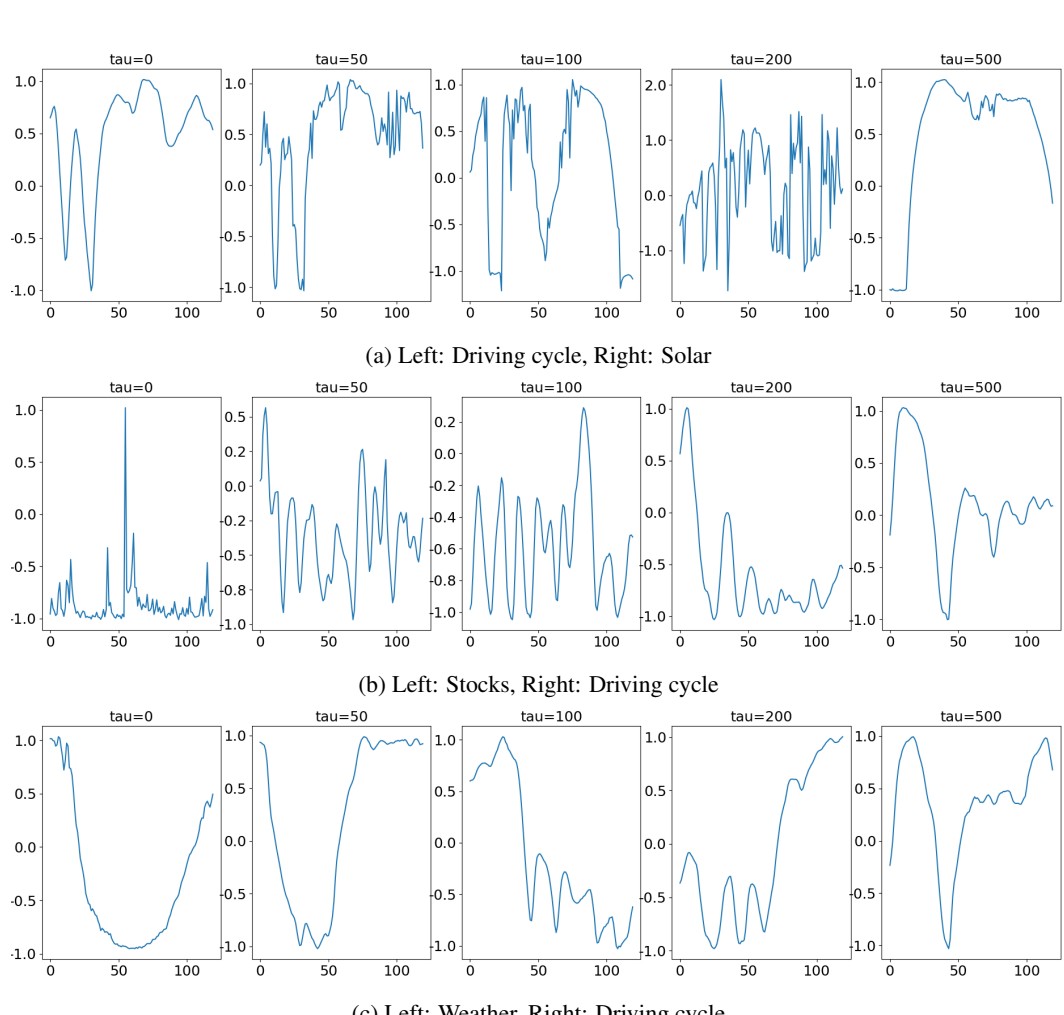

(a) Left: Driving cycle, Right: Solar

(b) Left: Stocks, Right: Driving cycle

(c) Left: Weather, Right: Driving cycle

Figure 13: Additional results for controllable conditions guidance. We fixed the random seed and generative fused data

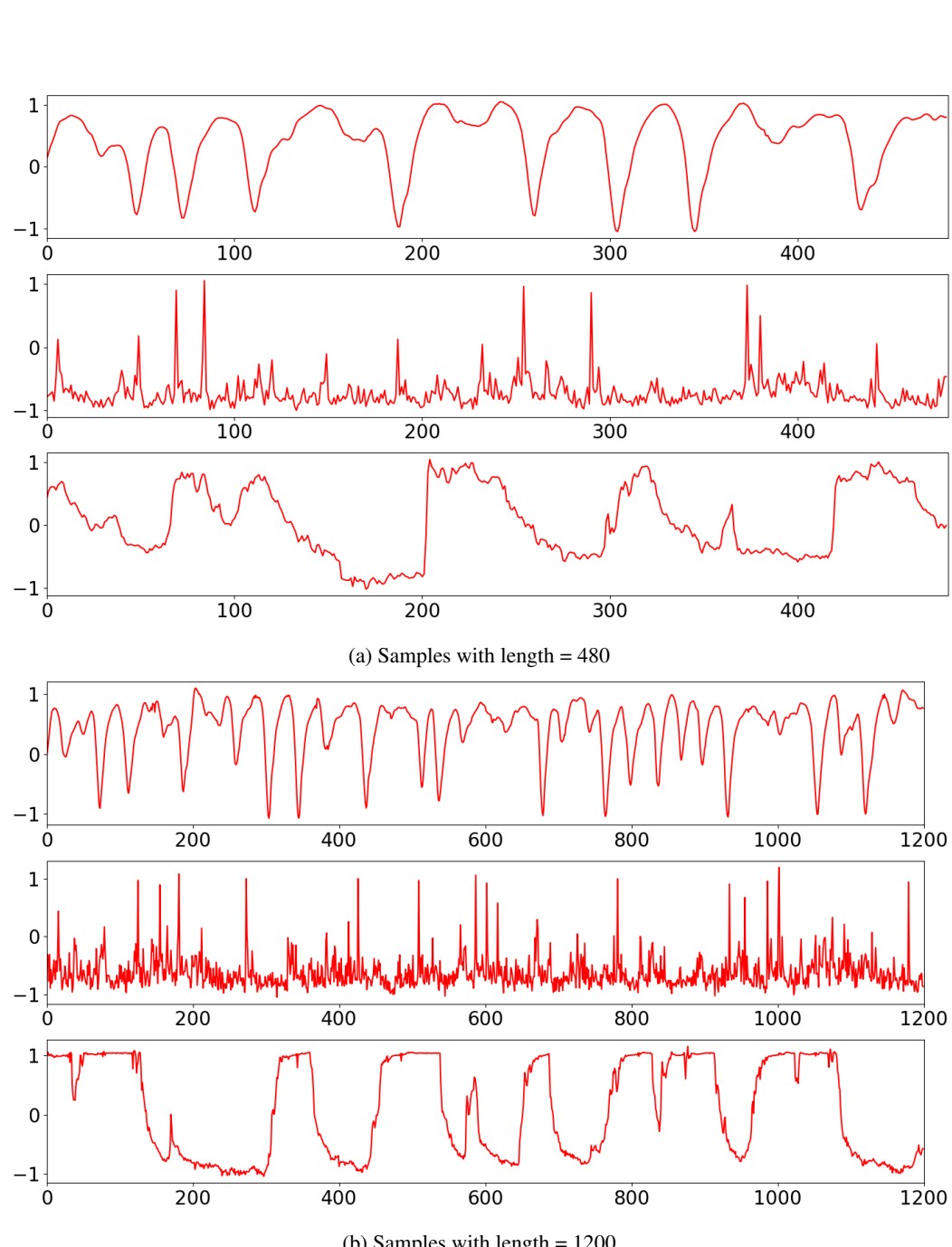

(a) Samples with length = 480

(b) Samples with length = 1200

Figure 14: From top to bottom: driving cycle, stock, weather

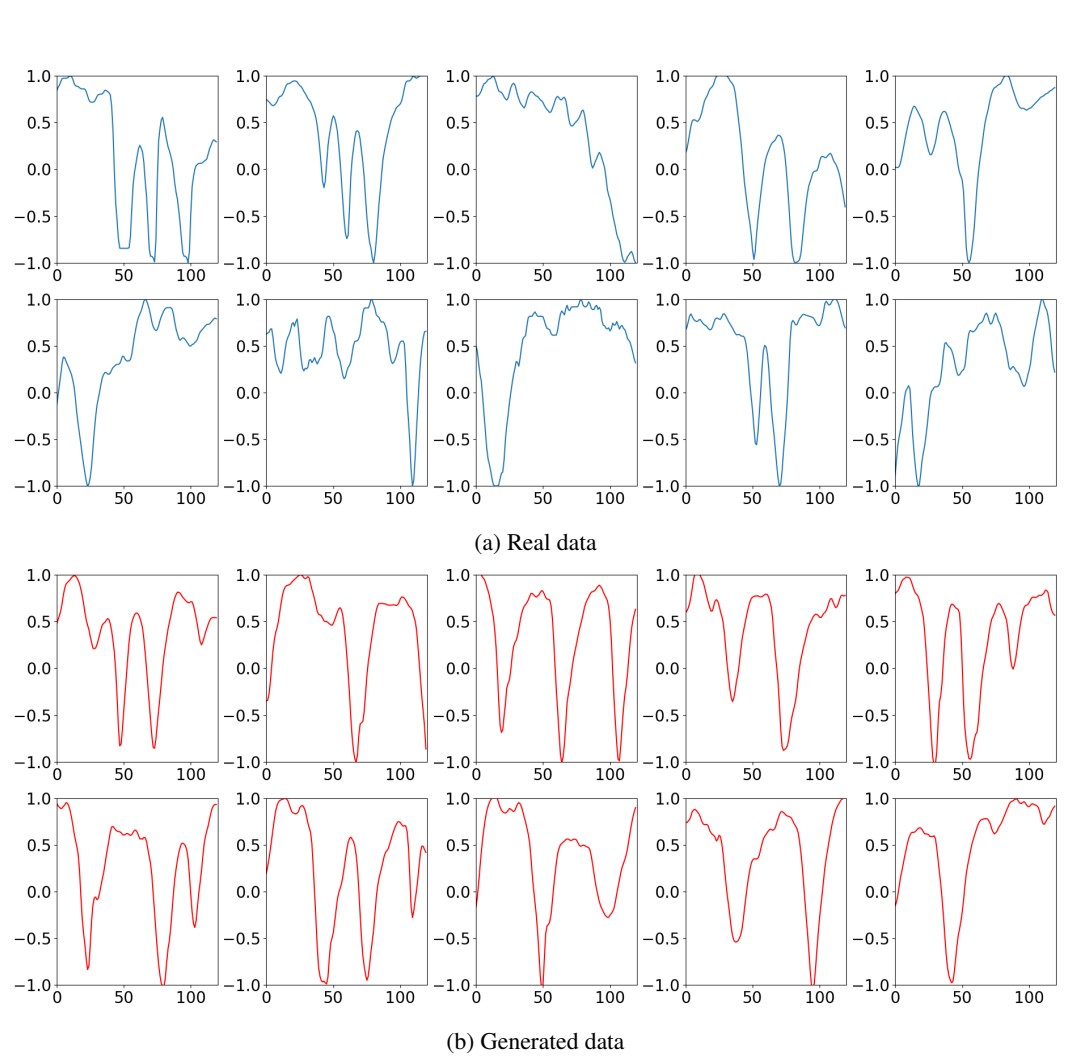

(a) Real data

(b) Generated data

Figure 15: Additional Dring cycle Generation

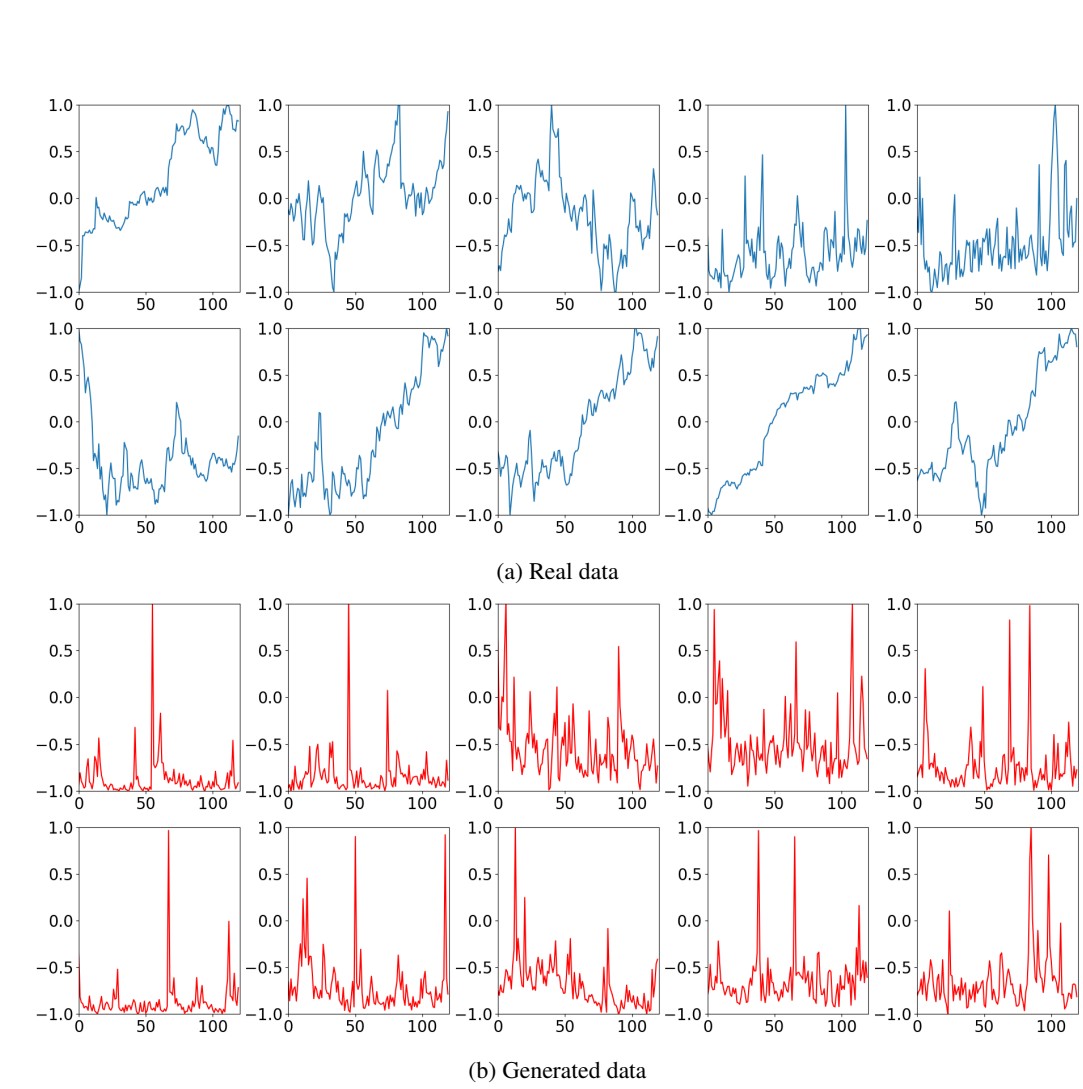

(a) Real data

(b) Generated data

Figure 16: Additional Stocks Generation

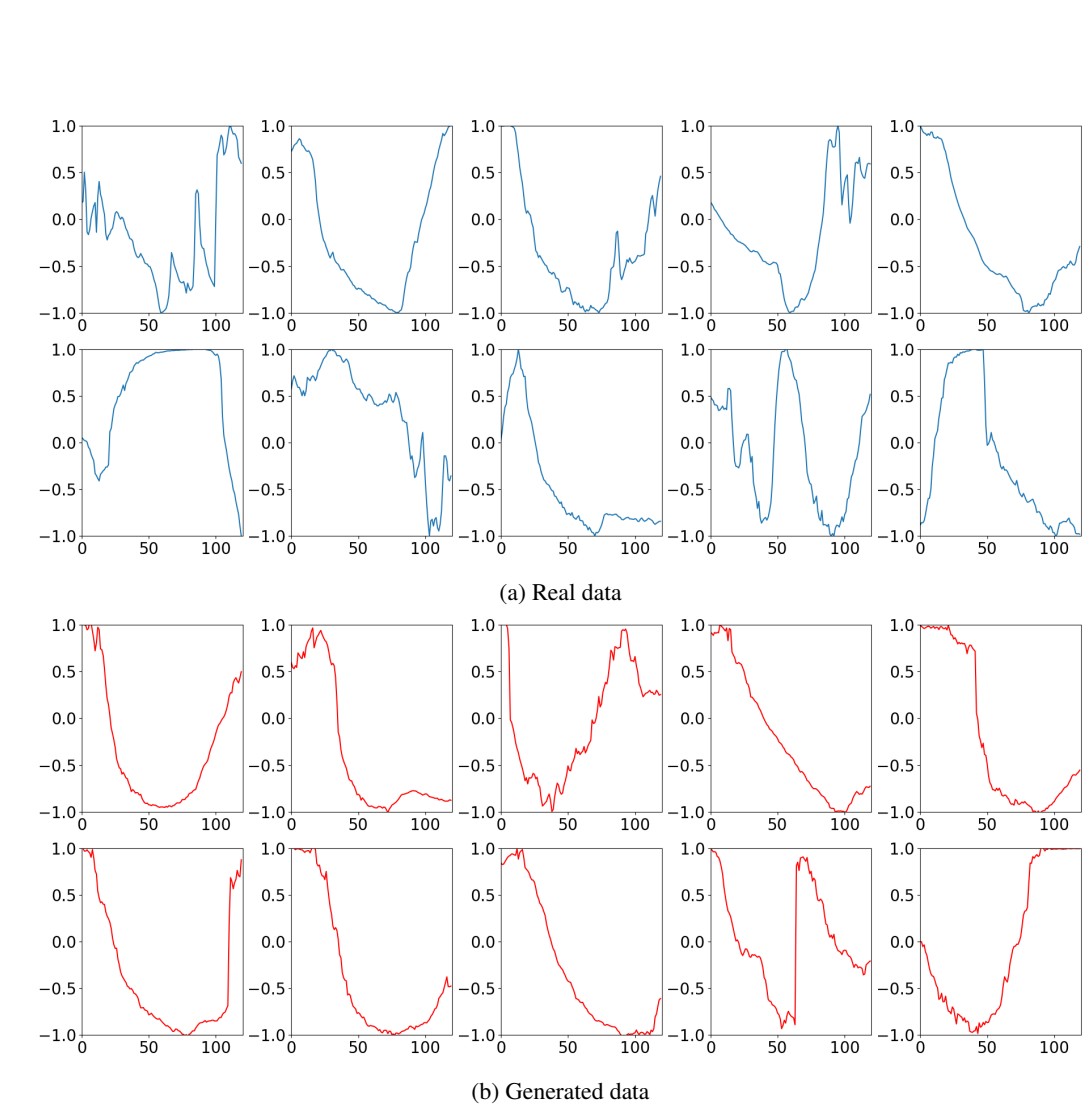

(a) Real data

(b) Generated data

Figure 17: Additional Weather Generation

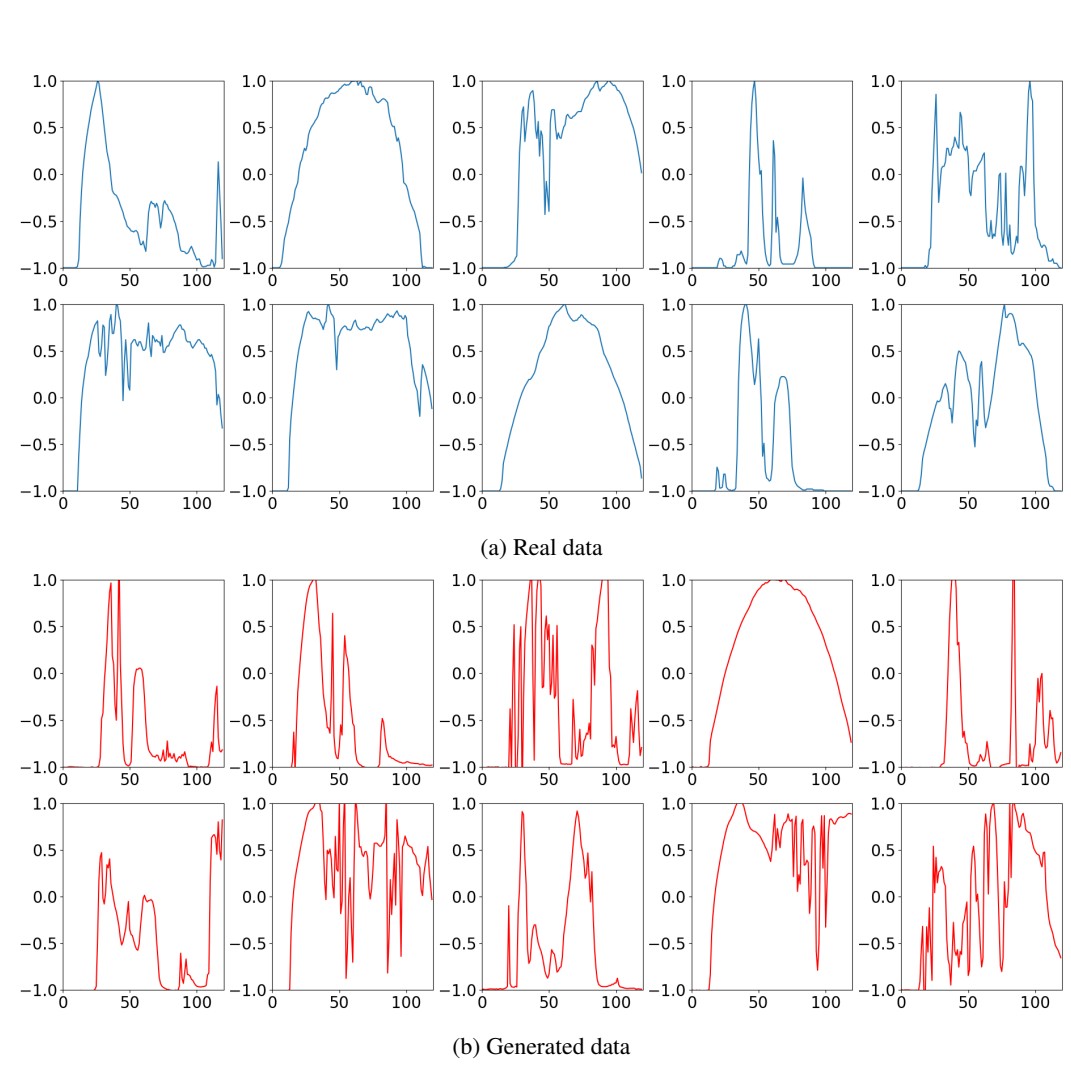

(a) Real data

(b) Generated data

Figure 18: Additional Solar Generation.

