# OpenReview forum: "High-quality and controllable time series generation with diffusion in transformers"
_ICLR.cc/2025/Conference — ICLR 2025 Conference Withdrawn Submission_

### Official Review · Reviewer_SKpk · 2024-10-29

**Soundness:** 2
**Presentation:** 3
**Contribution:** 2
**Rating:** 3
**Confidence:** 4

**Summary:**

This paper proposes a diffusion in transformer for time series generation, called timeDiT. There are a couple of architectural improvements such as dilated causal convolution to model the time-series data better. The smooth control sampling is introduced in order to fuse different categories in the diffusion steps. The experimental results are provided to show the effectiveness of the proposed approach in time-series generation tasks.

**Strengths:**

This paper incorporates the state-of-the-art techniques and methods proven by other domains such as image generation. Some sensible modifications are applied to make the system perform well for time-series data. A variety of time-series data is used in the experiments.

**Weaknesses:**

This paper should be rejected since 1) it is difficult to understand in the experiments 2) there are large gaps between the experimental results and statements derived from the results. The main arguments are listed in Questions below.

**Questions:**

1.	4.2 Unconditional generation: It is not clear how the proposed system is configured. Does “unconditional” mean that the samples are generated with y=y’=null?
2.	4.2 Conditional generation: Similar to the above question, how the system is set up for conditional generation? What is the y and y’, and what value is tau for these experiments?
3.	Table2: What are reasons to use Precision/Recall instead of Discriminative/Predictive Scores shown in Table1? How the classifier is trained, binary or multi-class detection, what data is used to train the classifier models?
5.	Table2: It is not straightforward to compare Table1 and Table2. FID is the common metric for both tables, but other metrics are different. Even for FID, there is degradation on some dataset (e.g. Stock: 9.06 -> 9.43, Weather: 6.09 -> 7.37), how should this be interpreted?
6.	4.2 Correlation constraints on multivariate sequences: Like earlier questions, it is not clear how the system is configured (y, y’, tau, etc.) for the multivariate task.
7.	Table 4: What do FID_a and FID_b represent?
8.	4.3 Longer sequence generation: It is claimed that “extended sequences have slight distortions in small segments but outperform the baseline”. It is not agreeable that the FID drop from 3.44 (T=120) to 37.18 (1200) is considered as “slight distortion”. In addition, the baseline result cannot be found to compare.

Things to improve the paper that did not impact the score:
1.	Point-wise layer: If the layer is something new, the explanation should be made. Otherwise, some reference should be added.
2.	Figure 2(b): What is x-axis and the size of circle? What are single and mixed pattern? Some descriptions are helpful.

---

> ### Author Response · Authors · 2024-11-13
> **Response to Reviewer Comments, thank you so much for your carefully reading!!**
>
> 1, 2.  The setup for conditional generation is to use a diffusion model for condition-guided generation. Specifically, in a mixed data training setting, the input is the class index, and the output is the generated data for that class. Since I planned to open-source the code, I did not elaborate too much on these details. There seems to be no open work for conditional generation for time series. Given that similar works have not provided condition-guided open-source benchmarks, we used accuracy and recall to measure the effectiveness of generating conditions, and employed FID to quantify the fit between the generated data and the real data within the modal distribution of the specific class. For instance, the FID for Solar is calculated by applying a feature extractor to both the generated Solar data and the real Solar data before computing the FID score.
>
> 3.  The precision and recall here refer to the accuracy of generating the specified conditions, rather than the discriminative prediction scores used to measure temporal characteristics and authenticity. We specifically need the former. The classifier is trained on original data; for detailed classifier usage, please refer to the citation in line L327
>
> 4. Table 1 presents results for training and generating within a single-class dataset, specifically focusing on generating data that aligns with that class. In contrast, Table 2 shows results on a mixed dataset, where conditional guidance is applied to specify and generate data for a particular class within the mix.
>
> 5. Here, y and y ′ represent different categories, and the style guidance aims to blend styles from different categories. This is useful in some downstream applications. For example, in driving behavior generation, y represents data from a cautious driver, while y ′ represents data from an aggressive driver. Similarly, in speech data, y could represent male voices, while y ′ represents female voices, with the goal being to edit the generated voice to be neutral, or to lean more male or female. Degree controlled by tau.
>
> 6. It connects with figure above, figure 5a, figure 5b.
>
> 7. According to the value among table 1,  the FID = 37.18 still preserves the curve features. The evaluation was based on averaging the FID scores from multiple random segmentations. Upon reflection, it seems this evaluation metric might lead to an inflated value because the translations caused by random segmentation could have a significant impact on the weather or solar data. Thank you for your helpful feedback.

---

> > ### Comment · Reviewer_SKpk · 2024-11-26
> > **Thank you for your response**
> >
> > 1.2.3 It is still not clear to me what "generating the condition" means. What does your diffusion model produce, e.g. index of the condition in addition to the data? If there is any additional output from the model, it should be clearly noted in the Fig. 1

---

### Official Review · Reviewer_e4YD · 2024-11-01

**Soundness:** 2
**Presentation:** 1
**Contribution:** 2
**Rating:** 3
**Confidence:** 4

**Summary:**

The paper adapts the DiT model to the time series domain with diluted casual convolutions. Furthermore, it promotes a new technique to smoothly control signal conditioning for time series generation. The paper offers a task of class conditioning and presents the method's capability in this task while also showing results on other unconditional generation tasks, including long-time series generation.

**Strengths:**

- The integration of causal dilated convolutions into the DiT for time series analysis is compelling, and the smooth guiding technique has demonstrated its effectiveness.
- I believe the author’s proposed task of class conditional generation is significant, and the ability to generate longer sequences is an important characteristic of any model and an interesting task.
- The model's efficiency compared to other diffusion models has been empirically validated.

**Weaknesses:**

- **Unconditional Generation Benchmark**
  - The benchmark lacks state-of-the-art methods like [1], which demonstrate significantly better results than those presented, leaving the claim of being state-of-the-art unsubstantiated.
  - The benchmark lacks datasets comparison such as MuJoCo and Energy (can be seen in Diffusion-TS benchmark).

- **Class Conditional Generation Benchmark**
  - The evaluation protocol is not clearly defined in the main text nor the appendix.
  - There is no baseline method provided.
  - It appears that the task has reached a saturation point, with nearly perfect scores.

- The multivariate task lacks clarity, and there is a notable absence of state-of-the-art comparison models [1].
- While the generation of longer sequences is a fascinating challenge, there are no baseline methods provided, though I believe a straightforward extension of current methods is feasible, and it is currently only evaluated on a single dataset.

[1] Generative Modeling of Regular and Irregular Time Series Data via Koopman VAEs

**Questions:**

- Could the author clarify what y and y' represent in the smooth control? It is unclear how they differ in the context of conditional classes. An example would be helpful.
- To enhance the related work section quality and keep it up to date, I recommend incorporating references to cutting-edge papers [2, 3, 4]. I emphasize that it’s not necessary to compare or include these in the benchmarks of this paper (experiments).

[1] Generative Modeling of Regular and Irregular Time Series Data via Koopman VAEs.

[2] Utilizing Image Transforms and Diffusion Models for Generative Modeling of Short and Long Time Series.

[3] SDformer: Similarity-driven Discrete Transformer For Time Series Generation.

[4] IDE: Frequency-Inflated Conditional Diffusion Model for Extreme-Aware Time Series Generation.

---

> ### Author Response · Authors · 2024-11-13
> **Response to Reviewer Comments**
>
> Dear Reviewer,
>
> 1. Here, y and y ′ represent different categories, and the style guidance aims to blend styles from different categories. This is useful in some downstream applications. For example, in driving behavior generation, y represents data from a cautious driver, while y ′ represents data from an aggressive driver. Similarly, in speech data, y could represent male voices, while y ′ represents female voices, with the goal being to edit the generated voice to be neutral, or to lean more male or female. Degree controlled by tau.
>
> 2. For weakness 1, the advantage of our model over DiffTS lies in its better fitting of diversity, as observed through the FID metric. The article analyzes that this advantage is due to the enhanced upper limit brought by DiT’s scalability. Furthermore, it is difficult to claim a ‘significant’ improvement at the lower bound. A sufficiently diverse dataset has already been employed, which is why we did not utilize custom MuJoCo data or the energy dataset that overlaps physically with solar data.
>
> 3. The setup for conditional generation is to use a diffusion model for condition-guided generation. Specifically, in a mixed data training setting, the input is the class index, and the output is the generated data for that class. Since I planned to open-source the code, I did not elaborate too much on these details. There seems to be no open work for conditional generation for time series. Given that similar works have not provided condition-guided open-source benchmarks, we used accuracy and recall to measure the effectiveness of generating conditions, and employed FID to quantify the fit between the generated data and the real data within the modal distribution of the specific class. For instance, the FID for Solar is calculated by applying a feature extractor to both the generated Solar data and the real Solar data before computing the FID score.
>
> 4. In this work, the model’s focus is not specifically on optimizing multivariate sequences. The objective of the experiments is to demonstrate the effectiveness of multivariate sequence generation, so reproducing and adding comparative experiments is not essential. Additionally, [1] does not appear to be open-source; if it were available, I would be happy to include it in comparisons. The long-sequence experiment similarly aims to demonstrate effectiveness by showing that the model can train on short data and sample long sequences. There is no comparable baseline available for this task.

---

> > ### Comment · Reviewer_e4YD · 2024-11-21
> > **Thank you for your response**
> >
> > MY concerns are still pending. I'll keep the original score.

---

### Official Review · Reviewer_BV8g · 2024-11-03

**Soundness:** 2
**Presentation:** 1
**Contribution:** 2
**Rating:** 3
**Confidence:** 4

**Summary:**

This work proposes TimeDiT, a diffusion transformer model that models temporal sequences. It is equipped with causal attention mask, dilated causal convolution and could generate a sequence longer than the training sequence in inference. It is also demonstrated that it's possible to exchange the condition during the backward process of the diffusion. The main evaluation is based on several metrics(IS,FID, etc.) on multiple datasets(solar, stock, weather, VED, Argoverse2). The proposed TimeDiT performs the best in the main evaluation. There are also other extended experiments demonstrating long sequence generation, smooth controllability and the ablation of several design choices in network architecture.

**Strengths:**

This work adapts DiT to model several temporal sequences, proposed several techniques(the use of zero-initialized AdaLN, smooth prior, causal dilated convolution) to improve the prediction/generation quality and described some issues observed when applying DiT to these datasets. Specifically, replacing time positional embedding with dilated convolution could be helpful in some applications.

**Weaknesses:**

Unfortunately, many detail of on the implementation and experiment is missing in the main text, and not even included in appendices. Also, besides the main evaluation(Table 1), the rest of experiments cannot justify what they aimed to claim, some of them have no numerical results at all. Also, the cause of "noise" without using dilated convolution in DiT is not well investigated. Although removing positional embedding and replace it with dilated convolution improved the evaluation metrics, it is not known if such treatment is generalizable.

Detailed comments are listed below:

- Soft prior: There's no theoretical or sufficient heuristic evidence to say that the unwanted noise is caused by masked attention. In Figure 2(a), the observation is merely an example. It cannot really verify whether "noise" and "number of peaks" will be increased in general due to the temporal masking on attention. Although soft prior can mitigate this phenomenon, it could create another side effect that's not easily observable. For example. if there's no external label to indicate the time, the model will not be able to reflect the long-term temporal distribution change, for example, the trend of global warming (although it can be resolved by using "year" as a condition in generation).

 - It's already reported in DDPM's paper that predicting noise leads to better quality than predicting x_0. However, to argue that predicting variance is useful, it needs theoretical/heuristic results.

- Adding dilated causal convolution can encourage the model to attend more on modulated past information. However, this is also achievable by standard attention with more layers. Is there any ablation to show this is more efficient than adding extra attention layers, MLPs, or other methods?

- The experiment on long sequence generation is not convincing. First, a length of 1200 is not really long considering much longer sequences has been tackled in LMs, images and sounds. Second, there's no result showing that the long sequence generated by the model is reflecting the long-term trend in the original data.

- The design of the classifier used for IS/FID evaluation is not explained at all. Although in Appendix there is an experiment showing that a similar trend on relative quality can be observed when using 5 classifiers, most of important details are missing. For example, the NN architecture of these classifiers, how the classification tasks are designed, and how are they trained, etc. Above all, testing with only 2-5 classifier is not convincing, what if all of them missed some important property of the data?

- The experiment on feature-fused generation (replacing condition in the middle of backward diffusion) does not have numerical result. How does it justifies the usefulness of condition fuse? In appendix C2, the description regarding Fig 12-13 is also not convincing. Few selected examples without quantitive analysis cannot justify the whether the "controllable guidance" is working or not.

- What's the significance of combining several arbitrary datasets for multi-variate generation, if they do not have dependency with each other?

- Diffwave is a work in 2020, it's hard to say it is "most advanced". Furthermore, there are several models[1,2] developed for Argoverse dataset, it would be good to compare with these models using the same evaluation protocol.
[1] B. Varadarajan, A. Hefny, A. Srivastava, K. S. Refaat, N. Nayakanti, A. Cornman, K. Chen, B. Douillard, C. P. Lam, D. Anguelov, et al., “Multipath++: Efficient information fusion and trajectory aggregation for behavior prediction,” in 2022 International Conference on Robotics and Automation (ICRA).   IEEE, 2022, pp. 7814–7821.
[2] Y. Liu, J. Zhang, L. Fang, Q. Jiang, and B. Zhou, “Multimodal motion prediction with stacked transformers,” in Proceedings of the IEEE/CVF Conference on Computer Vision and Pattern Recognition, 2021, pp. 7577–7586.

- Some of the metrics are not well explained. For example, in Table 2 and 6, it seems recall and precision are evaluated based on some events defined for each datasets, but they're not explained clearly. This is the same for "discriminative score" and "predictive score" in Table 1.

- In Fig 7-11, TimeDiT's generated distribution under t-SNE and PCA projection is sometimes more distinct than DiffTS and TimeGAN. This contradicts to the results shown in the main text (Table 1).

- In Appendix C4, Fig 15-18, capable of generating 1D curves that looks similar to human does not mean the model can really model the data correctly based on events and conditions. Again, numerical result is needed.

Minor errors and concerns:
- Did not explicitly define what is "DiffTS". Although it can be inferred that it's "Diffusion-ts" mentioned in related works, still this impacts the readability.
- Missing dash, "-", in dataset URL, it should be "https://www.bgc-jena.mpg.de/wetter/weather_data.html". Also, datasets should be referenced in the main text.
- Appendix A referred in main text is actually Appendix B.
- L714: Missing right ")".

**Questions:**

- Following the concern on soft prior, it would be good to have previous works or experiments to justify that positional encoding is the cause of the noise. It can be argued that for some data, its context can be entirely dependent on the past context, and therefore using absolute positional encoding is not reasonable. Just to mention, there's another work that can replace absolute time positional encoding but also support long sequence generation at inference time, for example, the rotary position embedding[1], which has shown its effectiveness in other tasks.
[1] RoFormer: Enhanced Transformer with Rotary Position Embedding, Jianlin Su, Yu Lu, Shengfeng Pan, Ahmed Murtadha, Bo Wen, Yunfeng Liu, https://arxiv.org/abs/2104.09864

Some other questions:
- L66: what does the "scaling properties" mean here? Although later in Figure 4 it is said that TimeDiT's "scaling properties" is the reason why it outperform DiffTS in the later stage of training, but I wonder what is the actual definition of this?

- The explanation of "point-wise layer" is unclear. It is reasonable that it must be something independent of sequence length, but isn't that any convolutional layer with proper padding could also fit the purpose?

- Although dilated convolution layer allows parallelization by the factor of dilation size, but unless the attention is also masked in a dilated manner, the parallelized evaluation will not have the same result to the sequential evaluation. How this is tackled in this work?

- In L341, since the classification task used for IS score in this work is not disclosed, how could a reader know such score difference is due to the loss of a class? Can you explain us how the IS classifier is designed?

- L347: Why smoother is always better? I believe it is data dependent. For example, speech data is non-smooth in this sense.

- L370: Where I can find the condition generation setup? How could we know if the result is good or bad if no other subjects in comparison?

- L474: Here it claims the proposed model outperforms the baseline, but there's no baseline for comparison in Table 5? In other modalities such as audio or music generation, generating much longer sample at inference time while maintaining a reasonable segment-wise FID is not a problem. Also, how does the model used in FID evaluation is designed?? usually the model has a fixed context length, how do you configure this for fair evaluation on longer outputs?

- L493: For fair comparison, does the 1D CNN has the some context length as DCC?

- L693 What's the term "dimension" mean here ?

---

> ### Author Response · Authors · 2024-11-13
> **Thank you so much! I understand what is going wrong/unclear about the paper, and I will provide some clarifications.**
>
> 1. Position Encoding. I believe some misunderstandings arose due to my incorrect explanation. In fact, to incorporate time sequences, I designed multiple approaches (MASK, DCC), considering model capacity compression as much as possible. My first observation was that the generation quality of DCC was far better than MASK. This led me to question whether the model should be simplified as much as possible. When I discovered that DCC's receptive field already contained position information, I attempted to remove the position encoding and achieved better experimental results. These results are recorded in the ablation experiment table. Recently, in another piece of work, I reintroduced position encoding because, in high-dimensional sequence generation, not using position encoding caused poorer fitting of the real constraints between dimensions. However, in this paper, the performance without position encoding was better. When exploring why removing position encoding led to this result, I found visible differences in the printed curves. Therefore, I have included Figure 2a for further explanation. In fact, no experiment has proven that position encoding adds noise. I believe the reason for the better FID and image representation when position encoding is removed is that the model is cleaner. This is because diffusion models sample noise from N(0,1), and separating this noise from the encoding might require higher capacity. However, I have always aimed to compress the model's capacity. Therefore, the correct conclusion is that removing position encoding works under the condition that: 1. It is a diffusion model, and 2. The physical constraints between different dimensions are not strong in the 1D time sequence. Since these thoughts and experiments are not the main focus of the paper, I did not include them in the main body, which may have led to the narrative misunderstanding.
>
> 2. Scalability refers to the model's ability to adapt to increasingly complex generation tasks (e.g., larger datasets) by increasing its capacity. In the paper, we proposed a model that enhances its capacity by modifying parameters (increasing hidden size and number of blocks) and training steps. As a result, the convergence improves with increasing capacity. In other words, the model's performance improves early on with the increase in capacity. However, DiffTS, due to its assumption of temporal seasonality and periodicity, is unable to learn some specific details in the later stages of training as the steps increase. This is because the periodicity and seasonality learned for non-stationary data might overfit.
>
> 3. Parallelization. In my work, I used the encoder-only architecture of the Transformer for representation learning, and therefore, no masking from the decoder was employed for sequence generation. The generation part is accomplished using a diffusion model.
>
> 4. L341. Since the theoretical maximum value of IS is 4, and similar fits always approach this theoretical maximum, I speculated that an IS value close to 2 indicates mode collapse. I conducted experiments to verify this result. However, readers who did not review the validation experiments might have doubts, and I apologize for the lack of rigor in this regard. Fortunately, the key point here is the comparison of IS for generative quality, which did not cause any misunderstanding
>
> 5. L347. I apologize for the possible misuse of the term 'smoothing.' What I intended to express is the consistency of the generated data with the real data in terms of the first/second derivatives, or that the rate of change in the discrete data should follow the same distribution.
>
> 6. L370. The setup here is to use a diffusion model for condition-guided generation. Specifically, in a mixed data training setting, the input is the class index, and the output is the generated data for that class. Since I planned to open-source the code, I did not elaborate too much on these details. Given that similar works have not provided condition-guided open-source benchmarks, we used accuracy and recall to measure the effectiveness of generating conditions, and employed FID to quantify the fit between the generated data and the real data within the modal distribution of the specific class. For instance, the FID for Solar is calculated by applying a feature extractor to both the generated Solar data and the real Solar data before computing the FID score.
>
> 7. L474. The sentence here was incorrectly phrased, as it should convey the idea that the segmentation is 'within an acceptable range.' Thank you for pointing that out. Since the FID = 37.18 still preserves the curve features, the evaluation was based on averaging the FID scores from multiple random segmentations. Upon reflection, it seems this evaluation metric might lead to an inflated value because the translations caused by random segmentation could have a significant impact on the weather or solar data. Thank you for your helpful feedback.

---

> > ### Author Response · Authors · 2024-11-13
> > **More**
> >
> > 8. Yes.
> >
> > 9. L693. The term 'dimension' used here is incorrect. What I meant is that each 'category' is treated as a separate feature to make the generation task's modes more diverse. For example, in the case of weather, there are 'temperature' and 'humidity.' These are treated as separate variables, but they both belong to the 'weather' category. Thank you for pointing that out.

---

### Official Review · Reviewer_FnTY · 2024-11-04

**Soundness:** 2
**Presentation:** 3
**Contribution:** 2
**Rating:** 6
**Confidence:** 2

**Summary:**

The paper presents TimeDiT, a model that leverages Diffusion Transformers (DiT) for generating high-quality, controllable time series data. By incorporating dilated causal convolution and an innovative smooth guidance policy, the model extends DiT to handle complex time series. Experimental results indicate that TimeDiT achieves state-of-the-art performance across several metrics and is able to generate longer sequences from short training sequences. Key contributions include a novel diffusion-based time series generation framework, a method for fusing features during diffusion, and the application of classifier-based metrics to evaluate model quality and diversity.

**Strengths:**

The model addresses an important gap in time series generation by **adapting diffusion models**, which have primarily been used in image generation, to handle temporal data. The incorporation of temporal characteristics through dilated causal convolution is innovative, making TimeDiT a pioneering approach in this space.

The **experimental** design is rigorous, comparing TimeDiT against several benchmarks on multiple datasets with varied characteristics. The use of classifier-based metrics enhances the evaluation's robustness, and the inclusion of both univariate and multivariate tasks demonstrates the model’s adaptability.

The paper is generally clear and well-structured, with **detailed explanations** of the model design and experimental procedures.
The model’s ability to generate longer sequences and control the output’s features makes it significant for applications in fields that rely on time series data. Additionally, TimeDiT's scaling properties and ability to capture diverse temporal patterns position it as a meaningful contribution to the field.

**Weaknesses:**

The model’s performance with low-dimensional datasets is promising, but **high-dimensional time series** data may pose additional challenges, especially in cases where dependencies across dimensions are intricate. There is limited discussion on how TimeDiT would perform in such complex multivariate contexts.

A notable limitation is the high computational cost of TimeDiT due to the diffusion process and the need for long training times to achieve high-quality outputs. The paper does not examine the scalability of the method with **larger datasets**.

Although the authors present results on generating longer sequences, the paper lacks detailed experiments and evaluations on **training with the same longer sequences**. This gap makes it difficult to assess the model's accuracy on longer time series. Furthermore, the lack of a direct **comparison with state-of-the-art methods** trained using longer time series makes the model's performance in this regard unclear.

**Questions:**

Please refer to Weaknesses.

---

> ### Author Response · Authors · 2024-11-13
> **Response to Reviewer Comments**
>
> Thank you for your understand.
>
> 1. High-Dimensional. This indeed represents a general limitation in time-series generation. Discussion of high-dimensional data in this paper is limited, primarily because Table 3 still shows conclusions similar to those for single-dimensional generation, namely that fidelity and diversity fit well, with weaker performance on real constraints (MSE) compared to the diffusion baseline, but still stronger than GANs and VAEs. This remains consistent with the trade-off in diffusion models, which sacrifice computational resources for better performance. According to some recent experiments, inter-dimensional constraints in high-channel generation tasks indeed pose a limitation. However, with increased model capacity and more training steps, the powerful representational ability of transformers increasingly reveals clearer relational separation across dimensions. (Some experiments support this observation, though they are not discussed here as it is not the main focus of this paper.
>
> 2. Large Datasets. Your point is indeed insightful. My recent work has involved generating sequences on datasets with up to 144 channels and a length of 6720, where the scalability of the approach has become remarkably evident. Although the numerical loss has not shown significant updates, the physical properties of the data are increasingly well-fitted with higher capacity and more training steps. A major constraint, however, is the substantial increase in training time, requiring around two days on my 10GB 3080 GPU. My equipment has not yet allowed me to explore the scalability limits fully (i.e., I find that increasing model capacity continues to enhance performance, though the point at which this benefit plateaus is unknown). Non-transformer-based networks are entirely unable to achieve this level of fitting. Thus, even with the additional training resources required, the trade-off appears worthwhile. In conclusion, DiT’s scalability remains robust on larger datasets, albeit at the cost of higher computational demand, which is still within an acceptable range.
>
> 3. Long sequences. This model in this paper is the only one among similar studies capable of completing the task of training on shorter datasets and directly generating longer sequences. Therefore, there is a lack of comparable benchmarks. While there is an evaluation of the effectiveness for long sequences, I do believe that it lacks a more rigorous assessment of the generation quality. For example, whether any segment of the same long sequence corresponds to the same mode, and if not, how these segments are distributed

---

### Official Review · Reviewer_T1E3 · 2024-11-04

**Soundness:** 2
**Presentation:** 2
**Contribution:** 2
**Rating:** 6
**Confidence:** 3

**Summary:**

This paper presents TimeDiT (Diffusion in Transformers for time series), a model designed to generate high-quality and controllable time series data by employing a Transformer-based diffusion approach. To improve on the limitations of conventional time series generation methods, the authors incorporate dilated causal convolutions and a guidance policy for style control, enhancing the Transformer’s temporal representation and understanding.

The model demonstrates scalability through its ability to generate longer sequences from relatively limited training data. Experimental results indicate that TimeDiT achieves state-of-the-art performance on several benchmarks, surpassing traditional GAN- and VAE-based models in terms of temporal fidelity and diversity in generated sequences.

- Personally, I find the application of diffusion within a Transformer framework intriguing, especially for time series data, where both structure and variability are critical. However, further discussion on the computational trade-offs involved in this model would strengthen the paper and offer valuable insights for researchers considering similar architectures in their work. It is between weak reject and weak accept to me.

**Strengths:**

- Enhances the model's ability to capture temporal dependencies without traditional position encoding.

- Enables smooth and flexible generation of diverse styles within time series data.

- Allows generation of sequences significantly longer than the training set, a vital feature for many real-world applications.

**Weaknesses:**

1. The model requires high computational resources and long training durations to converge, with the paper noting that over 100K steps are necessary for effective training. This contrasts with GAN or VAE models that converge faster, raising potential accessibility issues for those without extensive computational infrastructure.

2. The authors adopt classifier-based metrics for evaluation, but these are highly contingent on classifier accuracy. The classifier imperfections could impact fidelity and diversity assessments, potentially misleading users in applications where these evaluations are critical.

3. The authors note a lack of unified time series datasets, which hinders consistent comparisons. Without a standardized benchmark, it becomes challenging to generalize the model’s performance to broader datasets or applications outside the selected ones.

4. While the model performs well on cleaner time series, it has a tendency to amplify noise in high-density datasets. This noise issue is not fully addressed by the dilated convolution approach, and it raises questions about the model's performance on heavily fluctuating real-world data. There are also some theoretical works [e.g., A] on the population density estimation and error bounds, the author(s) may consider to add it for future discussion.

5. Diffusion models are known for long sampling times, and TimeDiT is no exception. While the model provides high-quality outputs, the extended sampling time could be a bottleneck in time-sensitive applications where fast data generation is needed.

***
A. Voice2series: Reprogramming acoustic models for time series classification, ICML 2021

**Questions:**

1. Given the classifier dependency, how can the metrics be adapted to reduce classifier bias, especially for more diverse or noisy time series data?

2. The model’s optimal performance is achieved under specific hyperparameters. Could the authors clarify the robustness of TimeDiT to changes in parameters such as learning rate, layer depth, and batch size, and suggest guidelines for optimal parameter tuning?

3. Many real-world applications involve non-stationary data, such as traffic patterns and stock prices. Has TimeDiT been tested on non-stationary data, and if not, what adaptations might be needed?

4. Given the autoregressive advantage of Transformers in other domains, can the authors compare TimeDiT’s long-sequence generation capacity to that of an autoregressive Transformer-based method?

5. The model has been evaluated on driving, stock, weather, and solar data. Could the authors discuss TimeDiT’s transferability to unrelated domains like medical or social time series data?

---

> ### Author Response · Authors · 2024-11-13
> **Response to Reviewer Comments**
>
> Thank you for your comments, you figure out some future work and related idea that may enhance my work, I appreciate it.
> 1.  About the computational resources.  On one hand, this computational cost is an inherent trade-off in diffusion models: training requires randomly sampling t from 𝑇 = 1000 steps and conditioning the model on t for each training iteration. Intuitively, this means the number of training steps would be a thousand times higher than for non-diffusion models. However, due to parameter sharing, the actual multiplier in my dataset was approximately ten times. On the other hand, I used a 10GB 3080 GPU, and the training time was under 10 hours. Unlike the benchmark study, my dataset consists of 60,000 real-world entries without sliding-window processing, making this implementation particularly simple. While computational cost cannot be overlooked, the experiment highlights the performance gains afforded by this trade-off. The balance ultimately depends on the diversity requirements of downstream tasks
> 2. The classifier. The classifier functions as a feature extractor for calculating the Fréchet Inception Distance (FID) with the original data. While its accuracy can impact FID calculations, this study is concerned only with relative FID values rather than the absolute FID score. As discussed in the appendix, even a convergent but imperfect classifier consistently reflects the comparative quality of FID scores across different models.
> 3. The discussion on stationarity and autoregressive modeling highlights that handling non-stationary data may represent a potential avenue for future work, especially if the generated data is intended for reinforcement learning applications that might necessitate a more rigorous analysis. In the current dataset used in this paper, however, domains such as stock data and driving data are inherently non-stationary. The working assumption here is that if 'non-stationarity' is equated to partially observable conditions, then capturing diversity becomes essential. This implies that predictions for non-stationary data are probabilistic, and the generated outputs display multimodal characteristics. Experiments indicate that the combination of Transformers with diffusion models significantly surpasses GAN-based or other autoregressive models in capturing diversity, as the latter are prone to overfitting to a single mode and may overlook rare events within the dataset. The baseline model 'DiffTS' is an autoregressive Transformer inspired by Autoformer's decomposition of periodic and seasonal components. While it likely performs well for stationary data prediction, for non-stationary data and specifically for generation tasks, omitting autoregressive priors is a preferable choice.
> 4. I conducted new work on generating driving behavior sequences within the same scenario, demonstrating that transfer learning within this domain is feasible. For different domains, while theoretically achievable, the conditions for data transferability remain unclear; incorporating some human insights might prove beneficial. From an application perspective, one possible requirement is that the two datasets should be processed into similar structures. For example, pretraining on a custom, highly correlated dataset with features like (a, b, c) before transferring to trajectory data with (x, y, rad) might allow transfer learning to retain temporal characteristics while relearning the interdependent constraints among a, b, and c. This, however, remains a personal hypothesis.
> 5. Consistency comparison. This is a rather disappointing outcome. In the baseline studies, a small dataset (such as a short segment of stock data) was used, processed with a sliding window to form a dataset, which then converged rapidly. The evaluation was limited to temporal and discriminative characteristics, focusing only on optimizations down to two decimal points. I believe the dataset used here is overly simplistic and insufficient to ensure practical applicability. Additionally, the evaluation criteria for generative models lack metrics for diversity and authenticity. As a result, I invested time in identifying more complex datasets, segmenting them as reasonably as possible in a physical context, incorporating FID for diversity comparisons, and rewriting the baseline to replace datasets and adjust parameters accordingly. However, the outcome is that consistency comparison is lacking. I will reflect on my mistakes and improve the paper before the next submission.
> 6. Noise. Thank you for sharing—this is exactly what I was looking for. In our application, there are stricter requirements for noise control. In fact, the experimental results show a variance discrepancy in the second derivative compared to real data. Although a Gaussian filter could easily address this issue, I am inclined to seek optimization within the model itself. If you have any further references, I would greatly appreciate them.

---

> > ### Comment · Reviewer_T1E3 · 2024-12-02
> >
> > I have read the authors' response and will keep my original scores.
> >
> > Meanwhile, I agree with the other reviewers that the theoretical justification and unconditional generation are somewhat lacking.
> >
> > I believe this draft can be further improved in the future by analyzing the joint system behaviors with latent space measurements.

---

### Note · Authors · 2025-03-09

**Comment:**

We would like to formally withdraw our submission from ICLR 2025. After further consideration, we have decided to make substantial revisions and improvements to the manuscript before submitting it to a more suitable venue. We appreciate the feedback received during the review process and thank the organizers for their time and consideration.

Best regards,

**Withdrawal Confirmation:**

I have read and agree with the venue's withdrawal policy on behalf of myself and my co-authors.

---

### Meta-Review · Area_Chair_GuKG · 2024-12-14

**Metareview:**

The paper introduces TimeDiT, a diffusion-based Transformer model for time series generation, incorporating innovations like dilated causal convolution and smooth style guidance. It claims to achieve state-of-the-art performance in generating high-quality, controllable sequences and extending short-sequence training to longer outputs. Strengths include adapting diffusion to temporal data, showcasing promising multivariate sequence generation, and evaluating diverse datasets with comparisons to baseline methods. However, the paper suffers from significant weaknesses: unclear and incomplete experimental setups (e.g., conditional generation protocols), lack of comparisons with state-of-the-art methods, theoretical justifications that are underdeveloped, and omissions in evaluation details (e.g., classifier design, metric explanations). Computational inefficiency and noise handling were raised but inadequately addressed. While the approach is innovative, these weaknesses, particularly in experimental rigor and clarity, provide compelling reasons for recommending rejection at this stage.

**Additional Comments On Reviewer Discussion:**

During the rebuttal period, reviewers raised concerns about computational costs, classifier dependency in evaluation metrics, scalability, and clarity in experimental setups. While the authors clarified the trade-offs in computational demands and highlighted the advantages of their approach, including novel contributions like smooth guidance and long-sequence generation, reviewers remained concerned about the lack of rigorous comparisons with state-of-the-art methods, insufficient testing on high-dimensional datasets, and limited clarity in describing the methodology. Despite acknowledging the model’s innovative aspects, such as its scalability and ability to handle temporal diversity, reviewers like T1E3 and FnTY were unconvinced about theoretical justification and benchmarking, maintaining marginal acceptance scores. Others, including e4YD and SKpk, advocated rejection due to gaps in evaluation and unclear configurations. Ultimately, unresolved concerns about clarity, benchmarking, and scalability led to a rejection, with reviewers encouraging revisions to address these issues.

---

### Decision · Program_Chairs · 2025-01-22

Reject